# Identifying genetic variations in *emm*89 *Streptococcus pyogenes* associated with severe invasive infections

**Masayuki Ono[1,2], Masaya Yamaguchi[1,2,3,4,5]\*, Daisuke Motooka[3,6,7], Yujiro Hirose[1], Kotaro Higashi[1], Tomoko Sumitomo[1,8], Tohru Miyoshi-Akiyama[9], Rumi Okuno[10], Takahiro Yamaguchi[11], Ryuji Kawahara[11], Hitoshi Otsuka[12], Noriko Nakanishi[13], Yu Kazawa[14], Chikara Nakagawa[15], Ryo Yamaguchi[16], Hiroo Sakai[17], Yuko Matsumoto[18], Tadayoshi Ikebe[19], Shigetada Kawabata[1,4]**

[1]Department of Microbiology, Graduate School of Dentistry, The University of Osaka, Osaka, Japan; [2]Bioinformatics Research Unit, Graduate School of Dentistry, The University of Osaka, Osaka, Japan; [3]Bioinformatics Center, Research Institute for Microbial Diseases, The University of Osaka, Osaka, Japan; [4]Center for Infectious Diseases Education and Research, The University of Osaka, Osaka, Japan; [5]Laboratory of Microbial Informatics, Microbial Research Center for Health and Medicine, National Institutes of Biomedical Innovation, Health and Nutrition, Osaka, Japan; [6]NGS Core Facility, Research Institute for Microbial Diseases, The University of Osaka, Osaka, Japan; [7]Integrated Frontier Research for Medical Science Division, Institute for Open and Transdisciplinary Research Initiatives (OTRI), The University of Osaka, Osaka, Japan; [8]Department of Oral Microbiology, Graduate School of Biomedical Sciences, Tokushima University, Tokushima, Japan; [9]Pathogenic Microbe Laboratory, Department of Infectious Diseases, National Center for Global Health and Medicine, Tokyo, Japan; [10]Department of Microbiology, Tokyo Metropolitan Institute of Public Health, Tokyo, Japan; [11]Department of Bacteriology, Osaka Institute of Public Health, Osaka, Japan; [12]Department of Public Health Sciences, Yamaguchi Prefectural Institute of Public Health and Environment, Yamaguchi, Japan; [13]Department of Infectious Diseases, Kobe Institute of Health, Hyogo, Japan; [14]Fukushima Prefectural Institute for Public Health Ken-chu Branch Office, Fukushima, Japan; [15]Division of Microbiology, Kyoto City Institute of Health and Environmental Sciences, Kyoto, Japan; [16]Sapporo Public Health Office, Hokkaido, Japan; [17]Niigata City Institute of Public Health and the Environment, Niigata, Japan; [18]Microbiological Testing and Research Division, Yokohama City Institute of Public Health, Kanagawa, Japan; [19]Department of Bacteriology I, National Institute of Infectious Diseases, Tokyo, Japan

**\*For correspondence:** yamaguchi.masaya@nibn.go.jp

**Competing interest:** The authors declare that no competing interests exist.

**Preprint posted** 02 July 2024

**Sent for Review** 25 August 2024

**Reviewed preprint posted** 03 January 2025

**Reviewed preprint revised** 20 May 2025

**Version of Record published** 24 July 2025

## eLife Assessment

This study provides an **important** and timely analysis of invasive and non-invasive *Streptococcus pyogenes emm*89 isolates, which have become a dominant serotype in the past decade. Using genome sequencing of 311 strains from Japan and comparing them with 666 global strains, the authors present **compelling** evidence in support of the identification of genetic factors linked to the invasive phenotype of *emm*89. The findings are both theoretically and practically significant in medical microbiology.

**Abstract** *Streptococcus pyogenes* causes mild human infections as well as life-threatening invasive diseases. Since the mutations known to enhance virulence to date account for only half of the severe invasive infections, additional mechanisms/mutations need to be identified. Here, we conducted a genome-wide association study of *emm*89 *S. pyogenes* strains to comprehensively identify pathology-related bacterial genetic factors (single-nucleotide polymorphisms [SNPs], indels, genes, or k-mers). Japanese (*n* = 311) and global (*n* = 666) cohort studies of strains isolated from invasive or non-invasive infections revealed 17 and 1075 SNPs/indels and 2 and 169 genes, respectively, that displayed associations with invasiveness. We validated one of them, a non-invasiveness-related point mutation, *fhuB* T218C, by structure predictions and introducing it into a severe invasive strain and confirmed that the mutant showed slower growth in human blood. Thus, we report novel mechanisms that convert *emm*89 *S. pyogenes* to an invasive phenotype and a platform for establishing novel treatments and prevention strategies.

## Introduction

*Streptococcus pyogenes* is a human-restricted gram-positive pathogen associated with a wide spectrum of diseases. While *S. pyogenes* often causes non-invasive diseases, including pharyngitis and impetigo in children, it is also known as a 'flesh-eating bacterium' owing to its involvement in life-threatening invasive diseases, such as necrotizing fasciitis and streptococcal toxic shock syndrome (STSS) (*Craik and Hla, 2022*; *Brouwer et al., 2023*). In 2005, more than 0.6 million people were estimated to have invasive *S. pyogenes* infections (*Carapetis et al., 2005*), and the reported incidence of invasive *S. pyogenes* infections continues to increase globally (*Craik and Hla, 2022*). In cases of severe infection, rapid bacterial growth and profound metabolic acidosis necessitate urgent surgical inspection and extended debridement with empiric antibacterial chemotherapy (*Stevens et al., 2014*; *Stevens and Bryant, 2016*). However, even with proper treatment, the mortality rate of patients with STSS remains high, ranging from 23% to 81% (*Walker et al., 2014*). Moreover, although several protective vaccine candidates against *S. pyogenes* exist, no safe and effective commercial vaccine has yet been licensed for human use (*Brouwer et al., 2023*; *Yamaguchi et al., 2013*).

S. pyogenes has been classified into at least 240 *emm* types based on a hypervariable region sequence of the *emm* gene, which encodes the M protein. This hypervariable region of the M protein is responsible for type-specific antigenicity and binds with high affinity to C4b-binding protein, a major fluid phase inhibitor of the classical and lectin pathways of the complement system that confers resistance to opsonophagocytosis (*Bessen et al., 2018*). Since the mid to late 2000s, *emm*89 strains have been increasingly isolated from samples obtained from patients with invasive diseases, becoming one of the most frequently identified lineages in developed countries (*Ikebe et al., 2021*; *Fay et al., 2021*). For example, in the United Kingdom, no more than 10 cases of invasive diseases caused by *emm*89 *S. pyogenes* were reported annually in the 1990s. In the 2010s, however, approximately 150 cases were reported out of the 1000–1500 invasive cases caused by all *emm* types of *S. pyogenes* in each year (*Turner et al., 2015*). Similarly, in Japan, in the 2010s, there was an increase in the number of patients with *emm*89 *S. pyogenes*-induced severe invasive infections. In 2018, *emm*89 strains were isolated from 36% of patients diagnosed with *S. pyogenes*-induced severe invasive infections, following *emm*1 strains, which were isolated from 57% of patients (*Ikebe et al., 2021*).

S. pyogenes *emm*89 strains have been genetically sub-clustered into three clades according to the *nga* promoter region patterns and the presence/absence of the *hasABC* locus, which is responsible for hyaluronan capsule synthesis. Clade 3 is distinct from clades 1 and 2 in terms of two features: overexpression of virulence factors NAD glycohydrolase (NADase) and streptolysin O (SLO), owing to mutations in the promoter region of the *nga-ifs-slo* operon, and the lack of a hyaluronan capsule (*Turner et al., 2015*; *Zhu et al., 2015a*; *Zhu et al., 2015b*). Although clade 3 strains have frequently been isolated from invasive diseases, their numbers from non-invasive infections have also increased. We previously reported no difference in the isolation frequencies of clade 3 strains between invasive and non-invasive diseases, at least in Japan, and concluded that the mutations in clade 3 are not responsible for the gain of invasiveness (*Hirose et al., 2020*). Therefore, there must be other genetic features within the *emm*89 strains that determine their phenotypes.

During infections caused by *emm*1 *S. pyogenes*, mutations in the two-component system, CovR/S, promote high virulence (*Walker et al., 2007*). These mutations cause the upregulation of DNase,

hyaluronan capsule, IL-8 protease, C5a peptidase, streptokinase, NADase, SLO, and superantigen SpeA, as well as the downregulation of cysteine protease SpeB and streptolysin S (SLS) (*Walker et al., 2007*; *Sumby et al., 2006*). The resulting mutants can prevent neutrophil death and subsequently promote tissue destruction and systemic infections. Epidemiologically, Ikebe et al. reported that nonsense mutations in *covR* and/or *covS* are present in 46.3% of *S. pyogenes* strains isolated from severe invasive infections in Japan, but only 1.69% of isolates from non-severe ones (*Ikebe et al., 2010*). Moreover, these studies indicated that the *covR/S* mutation is not responsible for all invasive clinical strains. Thus, we hypothesize that other mechanisms are involved in the development of invasiveness.

In the present study, we aimed to explore novel hypervirulent mechanisms of *S. pyogenes* by performing a pan-genome-wide association study (pan-GWAS) on *S. pyogenes*. Our pan-GWAS focused on the *emm*89 lineage of *S. pyogenes* to detect lineage-specific factors and minimize false positives due to lineage differentiation. For a comprehensive analysis, we constructed the core genome and pan-genome of *emm*89 *S. pyogenes* strains and evaluated the effect of single-nucleotide polymorphisms (SNPs) on core gene alignment and accessory clusters of orthologous genes (COGs). In addition, we performed a k-mers-based GWAS to detect SNPs in the intergenic regions and multiple mutations. We collected and sequenced *emm*89 clinical strains isolated in Japan during 2016–2021, in addition to public *emm*89 genome sequences. Using these sequences, we investigated the bacterial factors associated with severe invasive infections in Japan and globally, using pan-GWAS. Based on the bacterial protein structural predictions, we then selected candidates with high potential relevance to the phenotype. Finally, we introduced an SNP related to non-invasiveness into a clinical strain isolated from a severe invasive infection and examined the alteration of the bacterial phenotype through an ex vivo infection assay.

## Results

### Collection of *emm*89 *S. pyogenes* clinical isolates in Japan and construction of cohorts

We collected clinical *S. pyogenes* strains isolated between 2016 and 2021 from patients with non-invasive and severe invasive infections in Japan. The Ministry of Health, Labour, and Welfare of Japan has defined the clinical criteria of severe invasive β-hemolytic streptococcal infections as STSS, based on the STSS 2010 Case Definition of the Centers for Disease Control and Prevention in the US, with minor modifications, including the addition of encompassing symptoms in the central nervous system (*Supplementary file 1, table S1*; *Ministry of Health, Labour and Welfare, 2025*; *Centers for Disease Control and Prevention, 2021*).

For the *emm*89 clinical isolates, we collected T serotype TB3264 and untypable strains, in addition to *emm* genotype-identified strains. T-typing is a serologically based approach that is often used as an alternative or supplement to *emm* typing. T-antigens are trypsin-resistant surface antigens exhibiting extensive antigenic diversity (*Ikebe et al., 2003*). Isolates of a given *emm* type frequently share the same T serotype pattern (*Beall et al., 1998*; *Johnson and Kaplan, 1993*). The T serotype TB3264 corresponds to the genotype *emm*89 or *emm*94 (*Ikebe et al., 2021*; *Katsukawa et al., 2002*). A total of 207 clinical isolates were collected with the cooperation of the National Institute of Infectious Diseases and 10 public health institutes nationwide (*Supplementary file 1, tables S2 and S3*). We performed draft genome sequencing of the strains and identified their *emm* types. In total, 150 of these were determined as *emm*89, followed by 24 and 19 strains as *emm*4 and *emm*12, respectively. To focus on the pathogenic mechanisms underlying severe invasive infections in the *emm*89 cohort, we used 150 *emm*89 strains for subsequent analyses (*Figure 1A*, *Supplementary file 1, tables S2 and S3*). We previously determined the draft genome sequences of 161 *emm*89 strains isolated in Japan between 2011 and 2019 and determined their phenotypes using the same criteria (*Supplementary file 1, table S2*; *Hirose et al., 2020*). We combined these two sets and finally considered a total of 311 *emm*89 strains, including 135 severe invasive and 176 non-invasive isolates, as the Japanese cohort.

We also collected public genome sequences of *emm*89 *S. pyogenes* strains isolated from nine countries to further characterize the genetic properties of the Japanese cohort (*Supplementary file 1, table S2*; *Beres et al., 2017*; *Chochua et al., 2017*; *Davies et al., 2019*). In this study, the phenotypes of these strains were considered invasive if the diagnoses included severe infections, STSS,

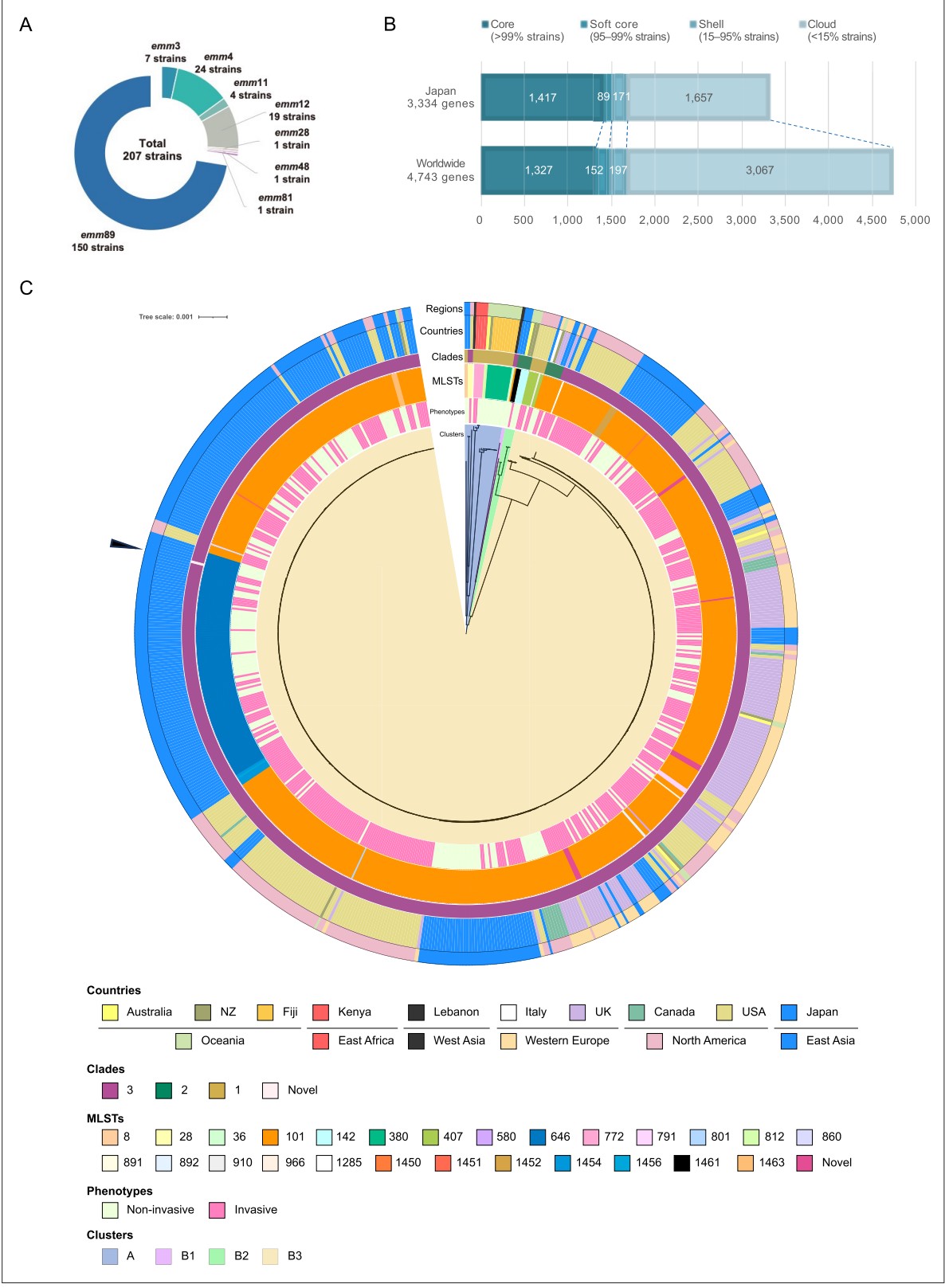

**Figure 1.** Pan-genomic and phylogenetic characterization of the collected *emm*89 *S. pyogenes* isolates. (**A**) *emm* genotyping of the 207 clinical isolates collected in Japan. (**B**) Pan-genome analysis of the Japanese and global cohorts. All genes detected in each cohort were classified into four groups, according to prevalence: core, soft-core, shell, and cloud genes. (**C**) Phylogenetic tree for the global cohort, based on the sequences of the core genes. From the inside, the color bars show clusters, phenotypes, MLSTs, clades, and countries where strains were isolated, and global regions, in the order

*Figure 1 continued on next page*

*Figure 1 continued*

mentioned. The root of the tree was set as the mid-point. The scale located upper left indicates 0.001 times substitution of the bases on average. Arrowhead indicates strain belonging to the novel clade. MLST, multilocus sequence typing.

The online version of this article includes the following figure supplement(s) for figure 1:

**Figure supplement 1.** Phylogenetic tree for the Japanese cohort.

**Figure supplement 2.** Variations in the *nga* promoter region.

**Figure supplement 3.** Pipeline for bacterial pan-genome analysis, pan-genome-wide association study (pan-GWAS), and GWAS constructed in this study.

invasive infections, necrotizing fasciitis, bacteremia, or sepsis, and isolation sites were described as normally sterile sites, such as the blood, brain, kidney, muscular tissue, or brain. Consequently, we identified 666 strains in the global cohort, including 420 isolates from invasive cases and 246 from non-invasive ones (*Supplementary file 1, table S2*).

## Pan-genome and phylogenetic analyses reveal both shared and distinct features in the Japanese and global cohorts

To determine the core genes and gene distribution in both cohorts, we performed pan-genome analyses. In the Japanese cohort, 1417 core genes common to more than 99% of all isolates were determined out of the 3334 different genes detected within the 311 strains. In contrast, the global cohort was more diverse, with 4743 different genes, of which 1327 were core genes (*Figure 1B*).

Next, we calculated the phylogenetic relationships based on the maximum likelihood of the core gene sequences (*Figure 1C*, *Figure 1—figure supplement 2*). The tree for the global cohort branched into four clusters, with clusters A, B1, B2, and B3. Cluster B3 included 640 genetically similar strains isolated mainly from Europe, North America, and Japan, whereas cluster A comprised 19 strains isolated from Oceanian countries, Kenya, Lebanon, the US, and Japan (*Figure 1C*). The phylogenetic tree for the Japanese cohort could also be clustered as in the case of the global cohort, with no significant difference in the proportions of strains classified into each sub-cluster (chi-square test, p = 0.13; *Figure 1—figure supplement 2* and *Supplementary file 1, table S4*). Thus, we concluded that the overall phylogenetic features of *emm*89 strains were distributed similarly in Japan and other areas, especially Europe and North America. Within cluster B3, we identified a non-invasive strain from Japan that had no identical pattern to the reported *nga* promoter variations (*Figure 1C*; *Turner et al., 2019*). This pattern is likely a subtype of clade 3 as it shares the haplotype $A_{-27}G_{-22}T_{-18}$, which is distinctive of clade 3, but has a mutation in the –10 box (*Figure 1—figure supplement 2*; *Turner et al., 2019*). Thus, we named this novel *nga* promoter variation type 3.4 (*Turner et al., 2019*). Multilocus sequence typing (MLST) analysis revealed that in cluster B3, 522 strains (80.9%) were ST101, and 96 strains (14.8%) were ST646 (*Figure 1C*). Notably, ST646 was the second most dominant type and a Japan-specific lineage. Moreover, they only differed in the 295th nucleotide in the *murI* locus, one of the seven loci that determine MLST, suggesting that both lineages have a genetically close relationship. Eight novel MLSTs were determined (ST1450, 1451, 1452, 1454, 1455, 1456, 1461, and 1463) and 15 novel MLST strains were detected in the Japanese cohort (*Figure 1—figure supplement 1*). Taken together, using phylogenetic approaches, we found that most strains from Japan and countries in Europe and North America share genetically close relationships, with only one unique lineage in Japan, ST646.

## Pan-GWASes detect SNPs/indels associated with invasiveness that are both common and specific to Japan and other countries

To discover all types of genetic variants in whole genes within *emm*89 *S. pyogenes* associated with (severe) invasiveness, we applied pan-genome analysis and performed three types of independent analyses, including pan-GWASes targeting SNPs in core genes and the presence or absence of all genes, and a GWAS targeting other variants located in intergenic regions spanning several nucleotides. The present pan-GWAS of bacteria within a single *emm* type minimized lineage effects, thus reducing false positives.

We extracted SNPs and single-nucleotide indels from core gene alignments and detected 24,627 and 47,060 SNPs/indels in the Japanese and global cohorts, respectively. Subsequent pan-GWASes

identified SNPs/indels associated with severe invasiveness in a Japanese cohort and invasiveness in a global cohort. To control for population bias, we calculated pairwise distance matrices and selected seven and three dimensions for the analyses of the Japanese and global cohorts, respectively (*Figure 2—figure supplement 1*). For each cohort, we performed a permutation test by conducting 1000 iterations of calculations with randomly permuted penotypes, with the significance level set at the 5th percentile of the 1000 minimal p-values (p = 5.75 × 10$^{-4}$ and p = 1.05 × 10$^{-4}$ for the Japanese and global cohorts, respectively). The pan-GWAS of the Japanese cohort detected 17 SNPs/indels in 13 core genes (*Figure 2A* and *Supplementary file 1, table S5*). Of the 17 significant variants, there were 7 single-nucleotide deletions (SNDs), 7 SNPs causing non-synonymous amino acid substitutions, and 3 SNPs causing synonymous substitutions. The *covS* gene (also known as *csrS*), encoding a sensor kinase of the two-component system CovR/S, contains four SNDs with the lowest p-values (p = 1.16 × 10$^{-7}$ for the 39th, 40th, and 46th nucleotides, and p = 1.15 × 10$^{-6}$ for the 125th nucleotide). These four deletions were associated with severe invasive infections.

We also performed a pan-GWAS for the global cohort and detected 1075 SNPs/indels significantly related to invasive infections among the 360 core genes (*Figure 2B, C* and *Supplementary file 1, table S6*). Among the significant SNPs/indels, 725 caused synonymous substitutions and 319 caused non-synonymous substitutions or frameshift mutations. Moreover, 19 SNPs induced nonsense mutations, whereas the effects of 12 SNPs/indels were unpredictable because of a lack of reference sequences (*Supplementary file 1, table S3*). Notably, 96 SNPs/indels accumulated in a single gene, *murJ*, which is involved in peptidoglycan biosynthesis, whereas 53 and 51 SNPs/indels were detected in *murE* and *group_1008*, respectively (*Figure 2—figure supplement 1*). The SNP with the lowest p-value (p = 1.35 × 10$^{-14}$) was *lacE*, which encodes the EIICB component of the lactose-specific phosphotransferase system (*Figure 2C*). This mutation was found to be associated with an invasive phenotype and mainly observed in strains isolated in the US. Compared with the significant 17 SNPs/indels in the Japanese cohort, 10 SNPs/indels were also detected in the global cohort, including 4 SNDs in *covS* and 1 SNP each in 6 loci (*Figure 2C*). Deletions at the *covS* locus were common among strains from several countries, including Japan. In contrast, SNPs in six loci, *gatA*, *group_1102*, *group_647*, *iscS_1*, *recU*, and *fhuB*, were present exclusively in Japan (*Figure 2C* and *Supplementary file 1, table S6*). These results suggest that several bacterial mechanisms cause severe invasive *S. pyogenes* infections, and some prevail worldwide, such as *covS* mutations, whereas others are specific to Japan.

## Pan-GWAS on COGs reveals 2 and 109 genes associated with severe invasiveness in the Japanese cohort and invasiveness in the global cohort, respectively

Next, we examined the associations of accessory COGs with severe invasiveness and global invasiveness. A permutation test determined significance levels as p-values of 1.09 × 10$^{-4}$ and 7.72 × 10$^{-5}$ for the Japanese and global cohorts, respectively. Two significant genes were detected in the pan-GWAS for the Japanese cohort: *group_184*, which encodes a hypothetical protein, and *divIC*, which encodes a septum formation initiator protein (p = 8.81 × 10$^{-6}$ and p = 6.72 × 10$^{-6}$, respectively; *Figure 3A* and *Supplementary file 1, table S7*). Although analysis of the global cohort revealed the presence of 169 genes that were significantly related to invasiveness, no genes were identical or homologous to the two genes detected in the Japanese cohort (*Figure 3B, C* and *Supplementary file 1, table S8*). Approximately 90% (152 of 169) of the significant COGs were found to be associated with non-invasiveness. Among the 169 genes, 37 encoded phage-related genes and 11 encoded mobile genetic elements (MGEs) such as transposase, integrase, and recombinase. The gene with the lowest p-value was *group_829*, which encodes a transposase and is related to the invasive phenotype (*Figure 3C* and *Supplementary file 1, table S8*). In addition, *group_2689*, which encodes a multidrug efflux transporter permease, *rhaR_3*, which encodes a transcriptional regulator involved in rhamnose metabolism, and *group_1829*, which encodes a vitamin B12 import transporter permease, were also substantially associated with the invasive phenotype. We identified several gene distribution patterns associated with invasiveness, suggesting that multiple independent genetic factors cause invasive infections (*Figure 3C*). On the other hand, the present pan-GWAS found no genes encoding known virulence factors significantly associated with invasiveness, thus further analysis of the relationships of detected distribution patterns with prophages and MGEs was performed. For calculating the pairwise correlation of the presence of significant COGs in the 666 strains, the COGs were clustered into

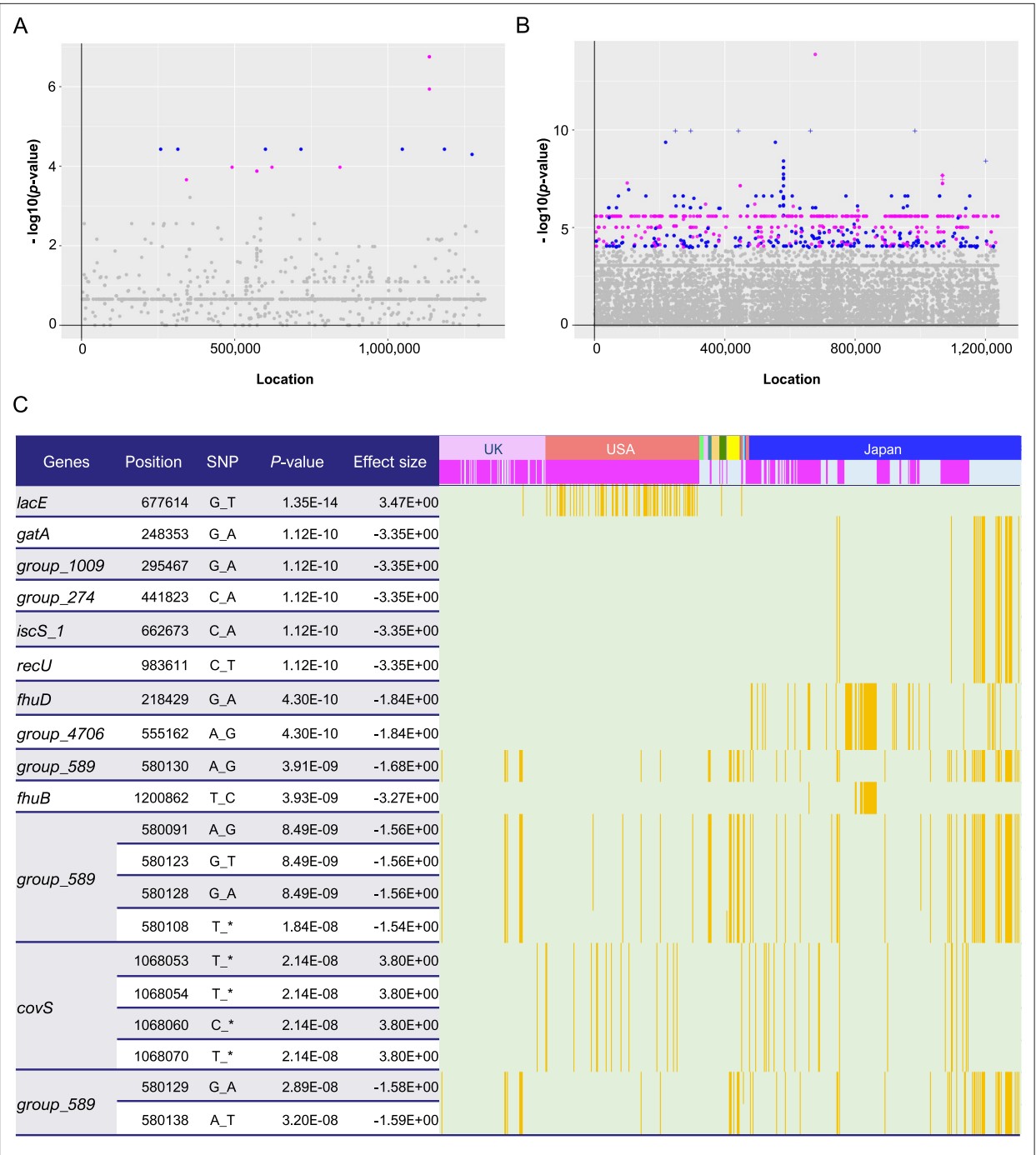

**Figure 2.** Pan-genome-wide association study on SNPs/indels. Manhattan plots for the Japanese (**A**) and global (**B**) cohorts. The *X*-axis shows the location of each SNP/indel on the core gene alignment, while the *Y*-axis indicates the p-value. For each cohort, a permutation test was performed by iterating the calculations 1000 times with randomly permuted phenotypes, with the significance level set at the 5th percentile of the 1000 minimal p-values (p = $5.75 \times 10^{-4}$ and p = $1.05 \times 10^{-4}$ for the Japanese and global cohorts, respectively). Plots with lower p-values than genome-wide significant levels are colored magenta and blue, based on the direction of their effect size (positive and negative, respectively). (**C**) Distribution heatmap for the global cohort, with the strains possessing the significant SNPs/indels colored orange. Only the 20 SNPs with the lowest p-values are shown in this heatmap. Colored bars above indicate countries and phenotypes, and magenta bars represent invasive phenotypes. Using the Roary program, gene names starting with 'Group_' were automatically assigned. Position indicates the location of each SNP/indel on the core gene alignment. The full results are shown in *Supplementary file 1, table S6*. SNP, single-nucleotide polymorphism; indel, insertion/deletion.

The online version of this article includes the following figure supplement(s) for figure 2:

**Figure supplement 1.** Pan-genome-wide association study (pan-GWAS) on single-nucleotide polymorphisms (SNPs)/indels.

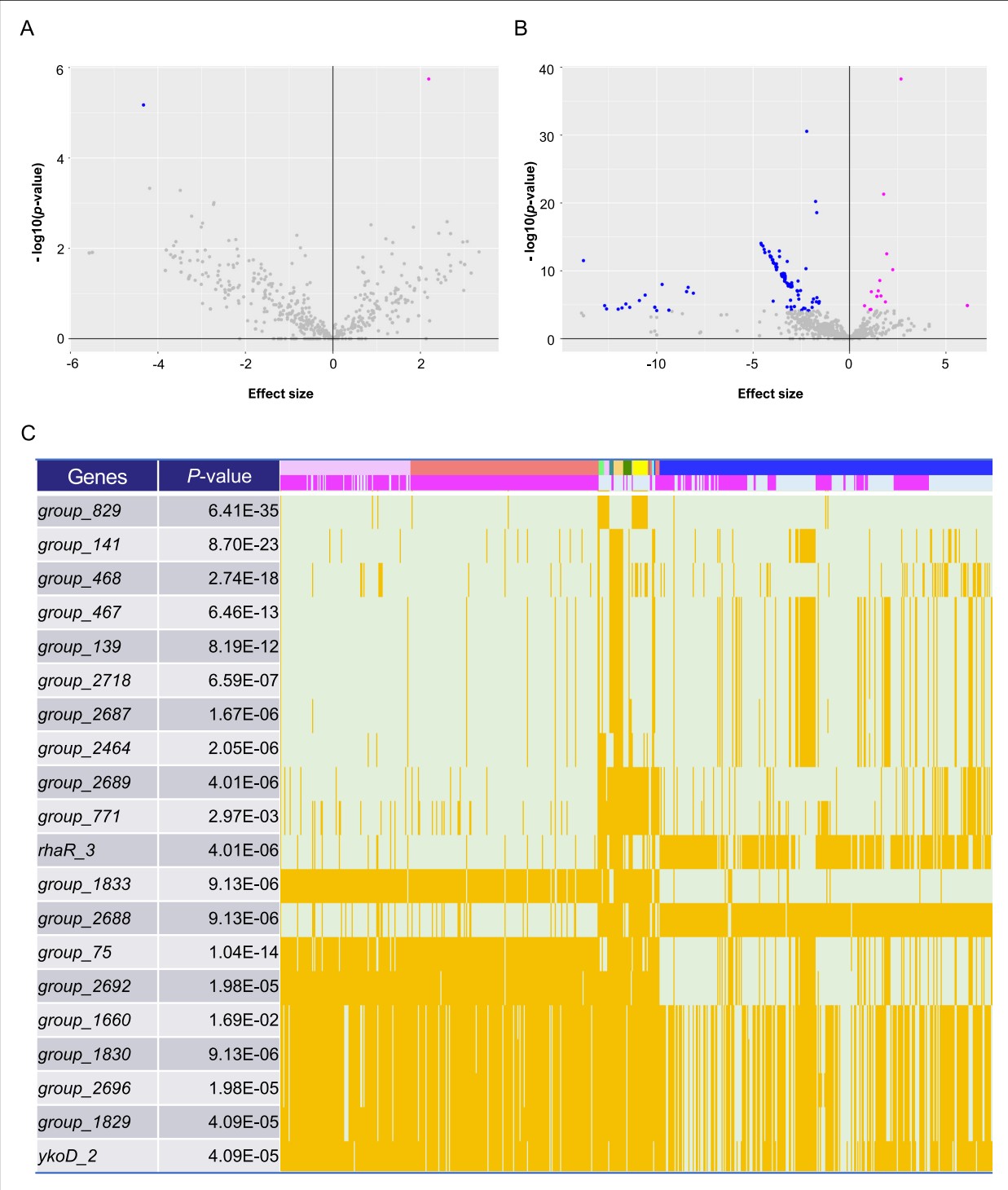

**Figure 3.** Pan-genome-wide association study on gene presence. Volcano plots for the Japanese (**A**) and global (**B**) cohorts. The *X*-axis shows the effect size, while the *Y*-axis indicates the p-value. Plots with a lower p-value than the genome-wide significant levels (p = 1.09 × 10⁻⁴ and p = 7.72×10⁻⁵ for the Japanese and global cohorts, respectively) have been colored magenta and blue, based on the direction of their effect size (positive and negative, respectively). (**C**) Distribution heatmap for the global cohort, with the strains possessing the significant genes colored orange. Only the 20 genes with the lowest p-values are shown in this heatmap. Colored bars above indicate countries and phenotypes, and magenta bars represent invasive phenotypes. Using the Roary program, gene names starting with 'Group_' were automatically assigned. The full results are shown in ***Supplementary file 1, table S8***.

The online version of this article includes the following figure supplement(s) for figure 3:

**Figure supplement 1.** Pairwise correlation of significant clusters of orthologous genes (COGs).

eight coexisting groups, seven of which contained phage- and/or MGE-related genes (*Figure 3—figure supplement 1*). The largest group comprised 65 genes including phage proteins, while the second largest with 42 genes was found to be associated with non-invasive infections and included *group_2689*, *group_1833*, and *ermA1*, encoding TetR/AcrR family transcriptional regulator, multi-drug efflux system permease protein, and rRNA adenine *N*-6-methyltransferase, respectively. Taken together, genes associated with invasiveness were found to encode mobile genetic factors and transporters, whereas major virulence factors were not significantly associated with invasiveness.

## K-mers-based GWAS detects both distinctive and identical variants compared to the SNP- and COG-based pan-GWASes

To detect SNPs/indels and multiple mutations in the entire genome, we extracted 31-nt-length k-mers from whole genomes and performed a GWAS. The k-mers-based GWAS can handle polymorphisms spanning more than one base, such as indels, inversions, and translocations, in both the coding and non-coding regions.

In the Japanese cohort, the k-mers-based GWAS detected two regions containing causative variants associated with severe invasiveness (*Supplementary file 1, table S9*). As shown in the de Bruijn graphs, overlapping k-mers were concatenated into a single node, and edges represent variability among the sequences (*Figure 4A–F*). The set of connected nodes and edges comprising each de Bruijn graph is called a complex. Nodes were determined to be significant if their *q*-value was less than 0.05. Significant nodes are shown in red and blue, indicating an association with severe invasive or non-invasive infections, respectively. The complex comprising the nodes with the lowest *q*-value ($q = 1.49 \times 10^{-2}$) was Comp_11 in the *covS* locus (*Supplementary file 1, table S9*). The causative mutations in *covS* were an SND and 10 nucleotide polymorphisms (*Figure 4A*). This deletion resulted in a frameshift mutation that shortened the length of CovS from 500 to 35 amino acids, leading to increased invasiveness, as previously reported in other *emm* types (*Cole et al., 2011*). Another complex significantly associated with severe invasiveness is Comp_2 ($q = 4.22 \times 10^{-2}$; *Supplementary file 1, table S9*). Comp_2 is a highly variable region containing eight hypothetical protein-coding genes, with high similarity within the first 75 bp. Significant k-mers were also mapped to the first 26 bp of *group_184* and 20 bp upstream (*Figure 4B*). These findings suggest that *group_184* possibly contributes to severe invasiveness through not only its presence but also by that of the upstream region.

Next, we analyzed the global cohort and identified mutations that were significantly associated with invasiveness in five regions (*Supplementary file 1, table S10*). The mutation with the lowest *q*-value ($q = 1.90 \times 10^{-2}$) existed in Comp_7 and was identical to the SND in *covS* present in the k-mers-based GWAS of the Japanese cohort. Thus, this SND is global, as are the four SNDs detected in the SNP/indel-based pan-GWAS.

Two significant k-mers were present in Comp_6, which were found to be an intergenic region of 270 bp ($q = 1.90 \times 10^{-2}$; *Supplementary file 1, table S10* and *Figure 4C*). We speculated that this region is related to regulation of gene expression. However, no promoter sequences were identified by utilizing MLDSPP, BacPP, and BLAST, thus the significance of this region remains to be clarified (*Paul et al., 2024*; *de Avila E Silva et al., 2011*). In Comp_24, with a high sequence variation containing *group_141*, *group_142*, and *group_143*, which encode transposases, the presence of a 281-bp sequence consisting of several k-mers was significantly associated with the invasive phenotype ($q = 1.90 \times 10^{-2}$; *Supplementary file 1, table S10* and *Figure 4D*). Furthermore, *group_141* was also identified in the COG-based pan-GWAS as a non-invasiveness-related gene along with *group_139* and *group_467*, which encode transposase and uncharacterized protein, respectively (*Supplementary file 1, table S8* and *Figure 3—figure supplement 1*). Taken together, the absence of an MGE containing *group_141* and the presence of another MGE harboring *group_142* and *group_143* may result in an invasive phenotype. In Comp_10, n27458, the *sagG* locus encoding the ATP-binding protein of the efflux transporter of SLS was significantly correlated with the non-invasive phenotype ($q = 2.40 \times 10^{-2}$; *Supplementary file 1, table S10* and *Figure 4E*). However, a significant SNP in n27458, *sagG* (A882G), was found to cause a synonymous mutation. The other significant mutation was present in the *fhuB* locus, encoding a putative ferrichrome transport system permease ($q = 2.40 \times 10^{-2}$; *Supplementary file 1, table S10*). This mutation was identical to SNP T218C detected in the SNP/indel-based pan-GWAS (*Figure 4F*). Moreover, this mutation changes the 73rd residue from valine to alanine in FhuB, which is a putative ferrichrome transport system permease.

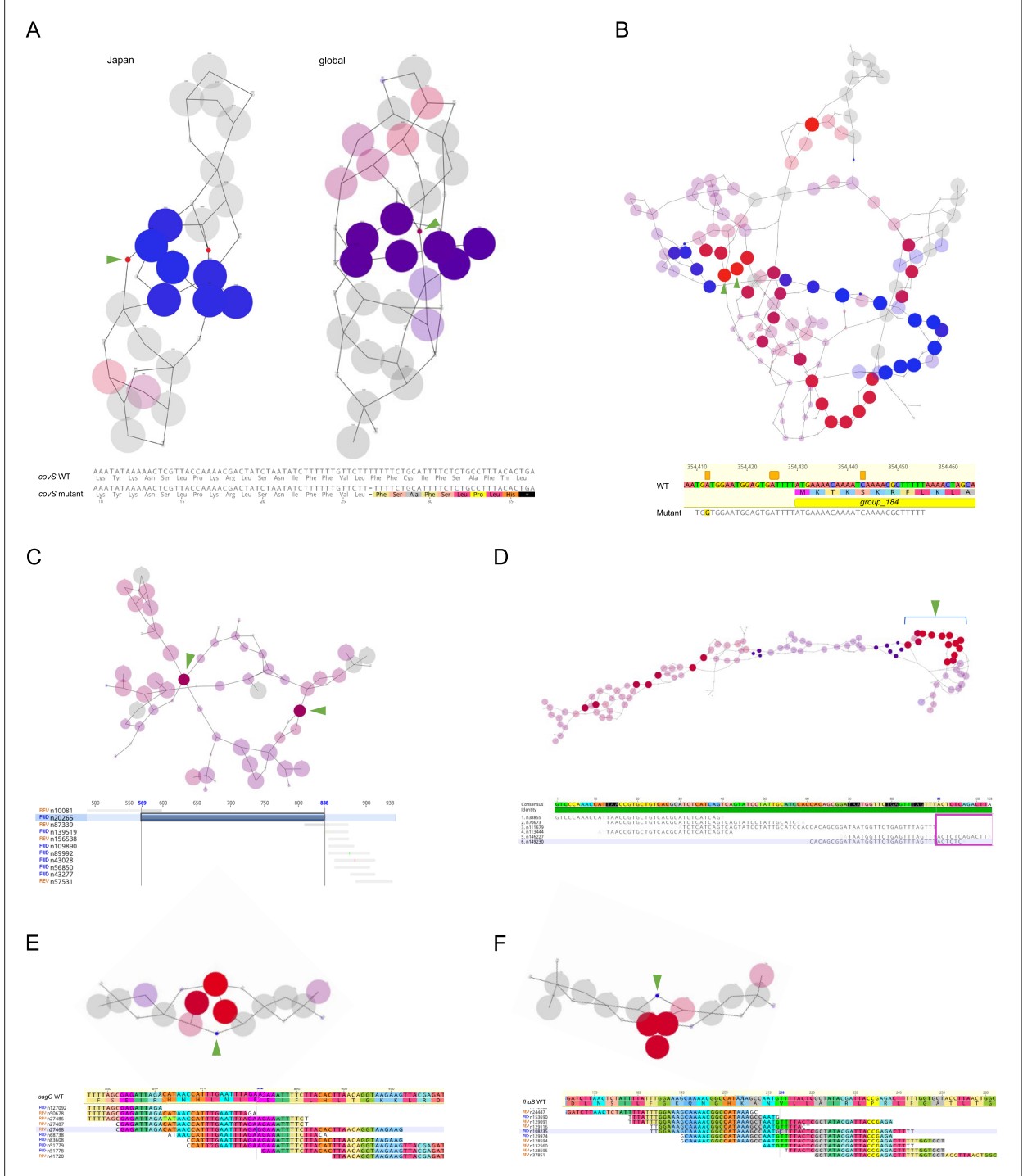

**Figure 4.** K-mers related to pathology. Detailed results of the GWAS on k-mers for the Japanese cohort in two genomic regions: *covS* (A, top left) and *group_184* (B); and the global cohort in five regions: *covS* (A, top right), an intergenic region (C), *group_141–143* (D), *sagG* (E), and *fhuB* (F). (A–F, top) de Bruijn graphs generated using DBGWAS. The respective nodes in the graphs indicate k-mers, with the significant nodes indicated using green arrowheads. The size of each node corresponds to the allele frequency. (A–F, bottom) Alignments of k-mers or maps of k-mers on the genome sequence of MGAS27061 around regions including significant mutations. Each base is colored according to the interpreted amino acids. (A) The arrowheads indicate significant k-mers that cause a frameshift mutation, truncating CovS protein to 35 amino acids. (B) The significant k-mers indicate that the presence of a sequence mapped on the first 26 bp of group_184 and its upstream 20 bp is related to the pathology. (C) Two significant k-mers indicate the same intergenic region of 270 bp. (D) The significant k-mers are indicated using arrowheads.

Therefore, the k-mer approach identified multiple variants, including the mutation identified in the SNP/indel-based pan-GWAS, *fhu*B SNP T218C. In addition, while the mutation detected in *covS* differed from that detected in the SNP/indel-based pan-GWAS, both caused frameshift mutations.

## AlphaFold-based prediction of the impact of the identified mutations on function

To assess the impact of mutations on protein function, we predicted the protein structure using Alpha-Fold (*Jumper et al., 2021*). Here, we present structural predictions for three representative proteins: LacE, whose mutation was observed mainly in invasive strains from the US; CovS, whose invasion-related deletions prevailed worldwide; and FhuB, which carries a prominent mutation in the Japanese cohort and is associated with non-invasiveness.

The invasive-related SNP in *lacE* substitutes the 554th glycine in LacE with valine. LacE was predicted to be a membrane protein with nine transmembrane regions (*Figure 5—figure supplement 1*). LacE is the EIICB component of the lactose-specific phosphotransferase system, and the EIIC and EIIB domains correspond to the transmembrane and intracellular regions, respectively. As shown in magenta in the model, the 554th residue is predicted to be in the intracellular EIIB domain (*Figure 5—figure supplement 1*).

Next, we predicted a homodimerized CovS model using AlphaFold, as the CovS of *S. pyogenes* forms homodimers (*Jain et al., 2020*). SOSUI predicted that CovS has two transmembrane regions (*Figure 5A*). Mutations detected in the SNP/indel- and k-mers-based GWASes were predicted to shorten the CovS protein to 35 and 45 amino acids, respectively. As the intracellular domain of CovS is in the C-terminal region and is involved in the phosphorylation of the transcriptional regulator CovR, frameshift mutations leading to CovS truncation would inactivate the protein, and thus, CovR function (*Figure 5B*).

As described above, the SNP T218C in the *fhuB* locus substitutes the 73rd valine of FhuB with alanine. FhuB is a component of an ATP-binding cassette transporter system that utilizes ferrichrome, which is one of the carriers of $Fe^{3+}$. FhuB is predicted to localize to the cell membrane and form a channel with FhuG (*Figure 5C*). SOSUI suggested that FhuB and FhuG are 9-transmembrane proteins. The FhuBG complex can bind to one molecule of the extracellular ferrichrome-binding lipoprotein, FhuD, and two molecules of the intracellular ATP-binding protein, FhuC. Therefore, we constructed a structural model of the FhuBCCDG complex (*Figure 5D*), which implied that the 73rd residue of FhuB exists in a region adjacent to FhuD. The mutation was predicted to induce formation of a small indentation on the molecular surface, thus increasing the surface area accessible to the solvent, and is considered to potentially affect the stability of the hydrophobic bond between FhuB and FhuD, and thus ferrichrome transport (*Figure 5E*). The SNP G538A in *fhuD* was Japan-specific, significantly related to severe invasiveness, and caused the V180I mutation in FhuD (*Figure 2—figure supplement 1*). The prediction suggested that the 180th residue is in an α-helix distant from the active site or the interactive sites with FhuB. Both valine and isoleucine are branched-chain amino acids, and the amino acid residue is located on a rigid structure, the α-helix (*Figure 5D*). Thus, we believe that this mutation is less likely to cause structural changes than the other non-synonymous mutations.

## The *fhuB* T218C mutation inhibits the growth of a severe invasive strain in human blood

Based on the pan-GWAS and GWAS results and predicted protein structures, we focused on the SNP *fhuB* T218C. We constructed a mutant strain, in which the SNP *fhuB* T218C was introduced, to further investigate its potential virulence. We selected the strain TK02, which carries the wild-type (WT) allele T218 in *fhuB* and was originally isolated from a sample obtained from a patient with severe invasive infection in Japan. We used a several times-passaged TK02 strain, TK02', as a WT strain and introduced the SNP *fhuB* T218C into it via allelic exchange mutagenesis with a thermo-sensitive shuttle vector. We then confirmed that there were no differences without the introduced mutation between the WT and *fhuB* T218C strains using whole-genome resequencing.

To reveal the effects of the SNP on invasiveness, we performed a transcriptomic analysis of the WT and *fhuB* T218C strains in THY broth and human blood. Principal component analysis revealed that the differences in the overall transcriptional profiles between the strains were more remarkable in blood than in THY (*Figure 6A, B* and *Supplementary file 1, tables S11–S14*). Among the

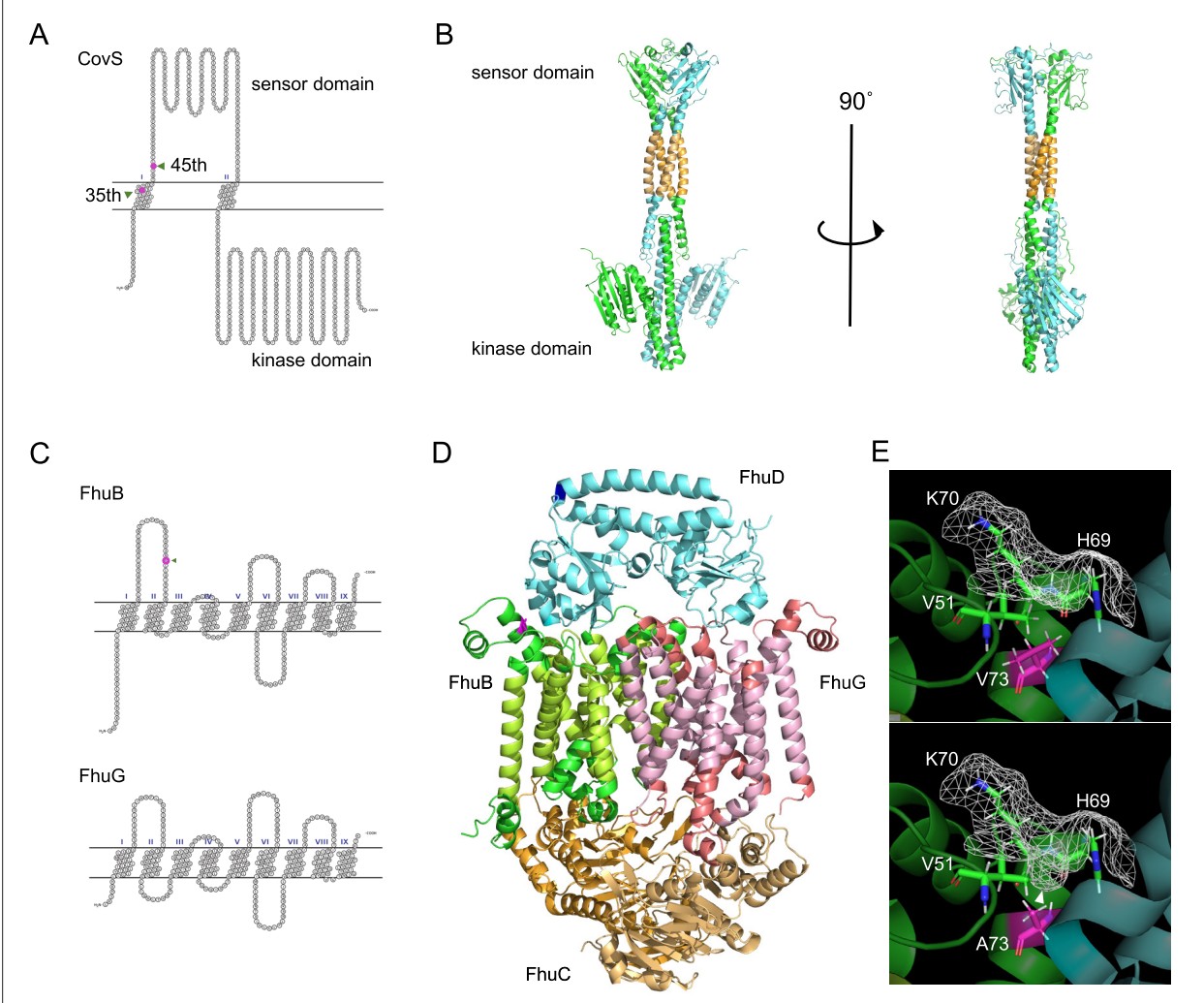

**Figure 5.** Predicted protein structure models. (**A**) Snake-like plot of transmembrane regions of CovS estimated using SOSUI. Frameshift mutations, detected by the single-nucleotide polymorphisms (SNPs)/indel- and k-mer-based GWASes, cause truncation of CovS at the indicated 35th and 45th residues, respectively. (**B**) Structural model of the CovS homodimer (ipTM + pTM = 0.614). The putative transmembrane regions are colored orange. The upper part is the sensor domain, while the lower is the C-terminal kinase domain involved in the phosphorylation of the transcriptional regulator CovR. (**C**) Snake-like plot depicting the transmembrane regions of FhuB and FhuG. The 73rd residue of FhuB is indicated by an arrowhead and in magenta. (**D**) Structural model of the FhuBDCCG complex (ipTM + pTM = 0.791). The putative transmembrane regions of FhuB and FhuG are colored green-yellow and peach, respectively. The upper part of the model is located in the extracellular region, while the lower part is in the cytoplasm. (**E**) The 73rd valine in FhuB, shown in magenta, was substituted with alanine. The molecular surface is illustrated with a wireframe, and that of the predicted indentation is shown with an arrowhead. ipTM + pTM: Weighted combination of interface-predicted TM and predicted TM scores. ipTM is used to measure structural accuracy in the protein–protein interface, while pTM is a metric for overall topological accuracy.

The online version of this article includes the following figure supplement(s) for figure 5:

**Figure supplement 1.** Predicted protein structure model of LacE.

overlapping genes, we found that the expression of CovR-regulating genes, including *speB*, *nga-ifs-slo* operon, and *sag* operon, was upregulated in the blood, compared to that in THY, in both the strains (***Figure 6C*** and ***Supplementary file 1, tables S12 and S13***). CovR has also been reported to regulate *sda1*, which plays a key role in invasive disease progression in *emm*1 strains; however, *sda1* does not exist in the *emm*89 reference strain MGAS27061 and WT strain (***Walker et al., 2007***). In human blood, the mutant strain resulted in the downregulation of *mga* and *emm* expression and upregulation of the expression of genes encoding virulence factors, such as *speC*, *scpA/B*, *endoS*, *fba*, *ska*, and *sfbX*, compared to those observed in the WT strain (***Supplementary file 1, table S14***). Although Mga regulates the expression of surface and secreted molecules important for colonization

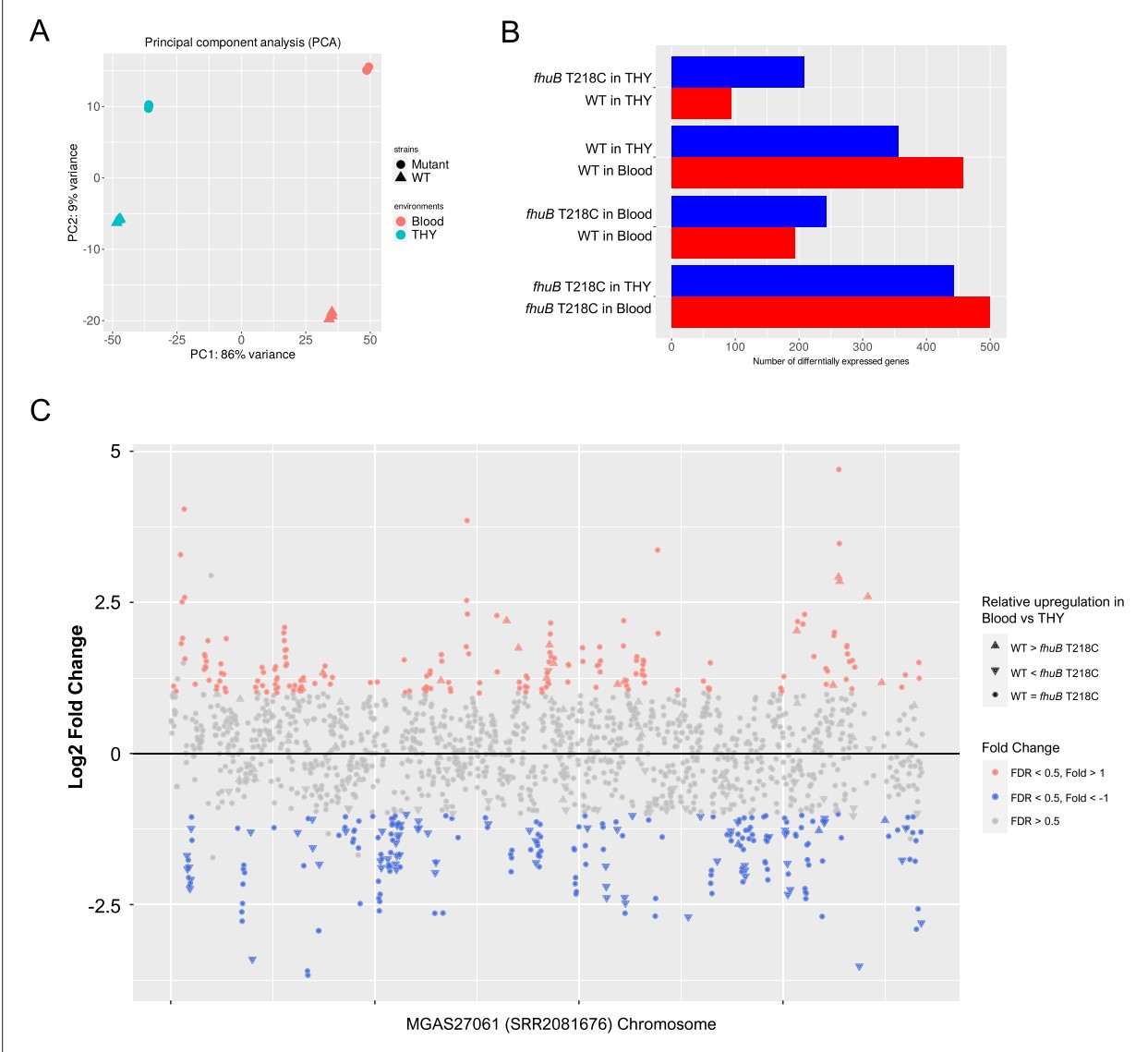

**Figure 6.** Transcriptome analysis of the *fhuB* T218C mutant strain in THY and human blood. (**A**) Principal component analysis plot of RNA-seq data. (**B**) Differentially expressed genes in four comparisons. (**C**) Plot of gene expression in the WT strain versus *fhuB* T218C mutant in human blood. Significantly up- and downregulated genes in the WT are colored red and blue, respectively. The shapes of the plots indicate relative transcriptional changes between THY and blood. Genes depicted with upward triangles are either significantly upregulated in the WT or downregulated in the mutant strain, in blood versus THY. The downward triangle plots indicate genes that are either downregulated in the WT or upregulated in the mutant strain, in blood versus THY. WT, wild-type; THY, Todd Hewitt broth supplemented with 0.2% yeast extract.

and immune evasion (***Ribardo and McIver, 2006***), no strong expression changes were observed in the Mga regulon, except in the *emm* gene. Notably, *fhuB*, *fhuC*, and *fhuD* were upregulated in the *fhuB* mutant in both human blood and THY (***Supplementary file 1, tables S11 and S14***). Moreover, the transcriptomic analysis revealed that both WT and *fhuB* T218C strains displayed upregulation of the *speB*, *nga-ifs-slo* operon, and *sag* operons in human blood. In addition, the *fhuB* T218C mutation caused the upregulation of *fhuBCD* expression. To determine whether *fhuB* T218C mutation affects iron transport, we measured the intracellular free ferric ion concentration in each strain and observed no differences among the strains in either environment (***Figure 7A***).

Next, to investigate the effects of SNP on bacterial survival in human blood, we performed a bactericidal assay using human blood. At 2 and 3 hr after incubation, the *fhuB* T218C mutant strain exhibited a significantly decreased survival rate than that of the WT strain (***Figure 7B***). To further determine the blood components that the attenuated survival of the mutant can be attributed to, we

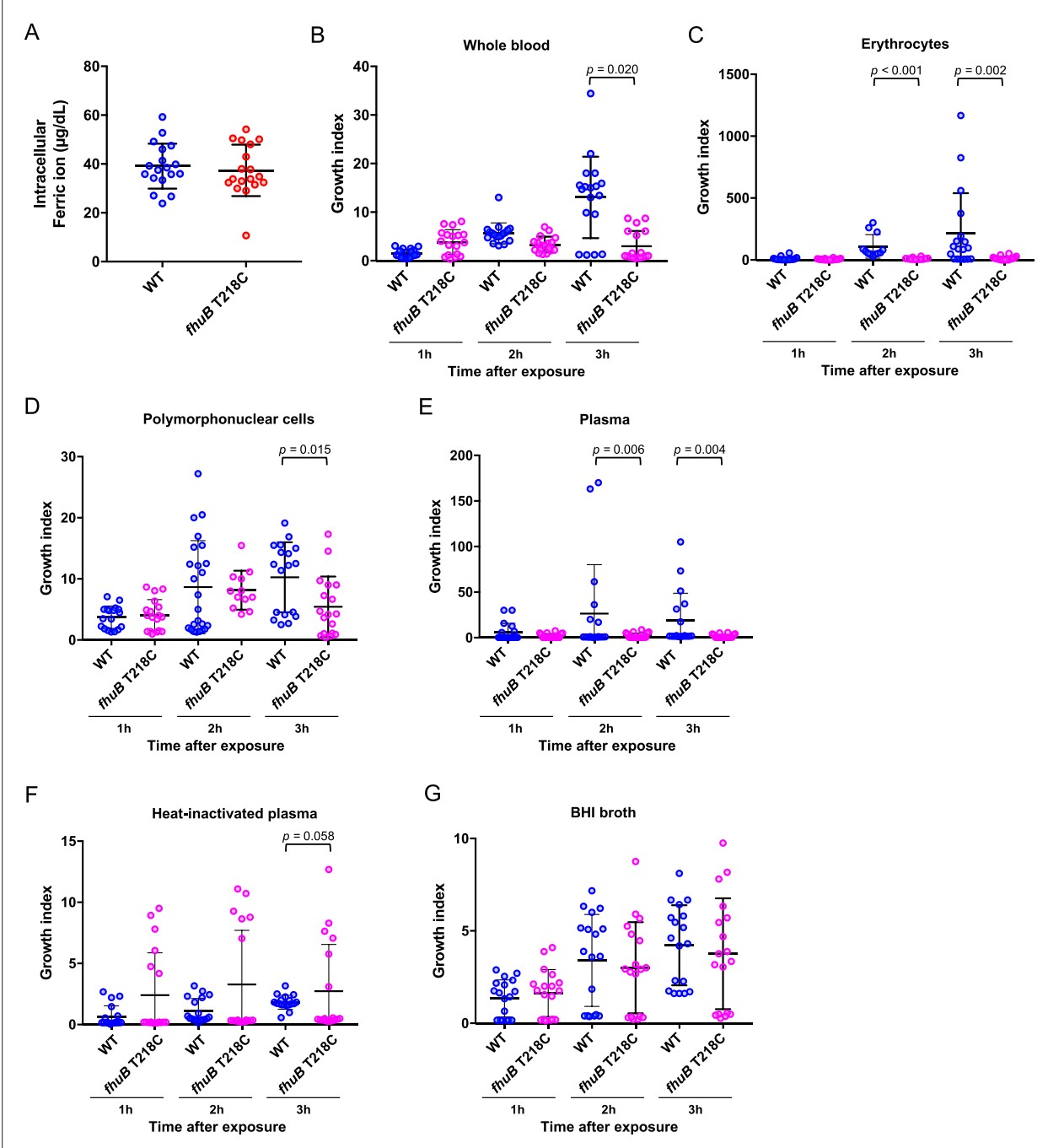

**Figure 7.** Effects of the *fhuB* T218C mutant strain on ferric ion uptake and bacterial survival in human blood. (**A**) Intracellular ferric ion assay. The wild-type (WT) and *fhuB* T218C mutant strains were incubated in healthy human blood for 3 hr, following which the intracellular ferric ion concentrations were measured. (**B**) Bactericidal assay in human blood. The WT and *fhuB* T218C mutant were mixed with healthy human blood to measure the ratio of bacterial counts at 1, 2, and 3 hr to those at 0 h after infection. Bactericidal assay in human erythrocyte-rich medium (**C**), polymorphonuclear cell-rich medium (**D**), plasma (**E**), and plasma inactivated by heating at 56°C for 30 min (**F**). (**G**) Bacterial growth in brain heart infusion broth. The data were pooled from three independent experiments, each performed in sextuplicate. Thick bars and error bars indicate means and quartiles, respectively. Statistical significance was determined using the Mann–Whitney test with Benjamini–Hochberg's correction.

compared bacterial survival rates in erythrocytes, polymorphonuclear cells, plasma, heat-inactivated plasma, and brain heart infusion broth (*Figure 7C–G*). Notably, after incubation with erythrocytes, polymorphonuclear cells, or plasma, the *fhuB* T218C mutant strain exhibited a significantly lower survival index than that of the WT strain, at 2 and 3 hr after incubation, as observed in whole blood

(*Figure 7C–E, G*). However, there were no significant differences between the survival rates of the WT and mutant strains in heat-inactivated plasma, suggesting that the mutant strain is susceptible to complement (*Figure 7F*). Taken together, the polymorphism T218C in *fhuB* impaired the survival of severe invasive strains in human blood through interactions with erythrocytes, polymorphonuclear cells, and complement.

## Discussion

The present study focused on *emm*89 *S. pyogenes*, known to cause increasing rates of invasive infections worldwide, and also assessed differences between *emm*89 strains causing invasive and non-invasive infections. By focusing on bacteria within closed phylogenies, false positives caused by confounding lineage effects were minimized, thus contributing to a higher level of accuracy of the pan-GWAS. We constructed a workflow for bacterial pan-GWAS and GWAS and explored the bacterial genetic factors related to severe or invasive infections to reveal the plausible mechanisms of pathogenesis. We independently performed pan-GWASes and GWASes for Japanese and global cohorts to investigate the associations of variants with strictly defined phenotypes in severe invasive *S. pyogenes* infections and with phenotypes in a broader context of invasive infections. Several GWASes have been conducted on *S. pyogenes* to date. Davies et al. extracted k-mers from the genomes of 1944 strains of *S. pyogenes* with broad *emm* genotypes and assessed their associations with invasiveness, and Kachroo et al. performed GWAS on k-mers using 442 *emm*28 strains isolated from either invasive or non-invasive infections (*Davies et al., 2019*, *Kachroo et al., 2019*). However, no studies have described the combined analysis of SNPs, genes, and k-mers exclusively using *emm*89 strains that share genetically close genomes and, thus, a large number of core genes. The investigations we performed led to the discovery of a tremendous number of causative variants, including not only SNPs in core genes but also accessory genes and insertions spanning intergenic regions.

Spontaneous mutations in *covR/S* genes potentiate the transition from localized to invasive infection by M1T1 *S. pyogenes* (*Cole et al., 2011*). We demonstrated that some SNDs in the *covS* locus were significantly related to the phenotypes in both the SNP/indel- and k-mers-based GWASes. CovS forms a two-component system with CovR that regulates the transcription of operons encoding virulence factors when phosphorylated by the histidine sensor kinase CovS (*Vega et al., 2016*). Sumby et al. suggested that systemic invasive infections are caused by the overexpression of virulence factors through the inactivation of CovS, as a result of the introduction of a depletive mutation in the *covS* locus after infection (*Walker et al., 2007*; *Sumby et al., 2006*). Several mutation sites causing such overexpression, such as the 7 nt insertion in *covS* of *emm*1 *S. pyogenes* and the insertion of adenine at the 877th nucleotide, have been reported (*Walker et al., 2007*; *Sumby et al., 2006*). Ikebe et al. demonstrated variations in the lengths of mutant CovS proteins by analyzing the gene sequences of 164 STSS strains in Japan (*Ikebe et al., 2010*). In line with their study, we found that the shortening of CovS that potentially occurred owing to frameshifts caused by SNDs is significantly associated with invasive infections, potentially contributing to the upregulation of virulence factors and invasiveness.

In contrast, we observed several factors associated with invasiveness to be independent of *covR/S* mutations. Based on conformational predictions, we selected SNPs with non-synonymous substitutions that were likely to affect protein function. Transcriptomic analysis findings suggested that the Japan-specific *fhuB* mutation associated with non-severe invasive infections contributes to the growth rate of *S. pyogenes* in human blood by adapting to the environment. In addition, *emm*89 clade 3 carries the identical promoter region pattern of the *nga* operon as *emm*1 strains, and the pattern conferred similarly high expression of *nga* and *slo* (*Turner et al., 2015*; *Zhu et al., 2015a*; *Turner et al., 2019*). Our RNA-seq data demonstrated that a severe invasive Japanese strain without *covS* mutations increased the expression of *speB*, *nga*, *slo*, and genes in the *sag* operon in the blood. Although *covR/S* mutations downregulate the expression of SpeB and SLS, SpeB and SLS act as virulence factors, allowing *S. pyogenes* to invade host tissues (*Terao et al., 2008*; *Sumitomo et al., 2011*; *Honda-Ogawa et al., 2013*; *Lin et al., 2009*). These data suggest that severe invasive infections have multiple gene expression profiles in addition to *covR/S* mutation-induced profiles within a single lineage, *emm*89, and that the synergy between optimizing bacterial survival in human blood and upregulating multiple virulence factors contributes to the development of severe invasive infections.

Two possible roles of the *FhuB* mutation in the pathogenesis of severe invasive infections are thus proposed. First, the mutation in *FhuB* could increase bacterial susceptibility to free radicals generated

in the presence of ferric ions in the blood. Generally, ferric ions are essential for the survival of almost all living organisms. Catalyzed by ferric ions, the Fenton and Haber–Weiss reactions generate hydroxyl radicals in erythrocytes (*Cimen, 2008*). Previously, we provided evidence that the iron in erythrocytes partially inhibits pneumococcal growth via a free radical-based mechanism (*Yamaguchi et al., 2013*). Although no significant difference was observed in the intracellular ferric ion concentration in the present study, the structural changes in FhuB may have caused an increased generation of free radicals in bacterial cells and the prevention of bacterial survival in human blood. Second, the *FhuB* mutation could affect bacterial transcriptional profiles and result in reduced fitness for survival in human blood. We also observed a lower survival rate of the mutant strain with polymorphonuclear cells (*Figure 7D*). Lower *emm* transcription in the mutant may lead to attenuated immune evasion. The M protein can bind C4-binding protein and factor H and inactivate the deposited C4b and C3b, leading to limited surface opsonization (*Laabei and Ermert, 2019*). Furthermore, complements could have inhibited the proliferation of the mutant strain, as suggested by the activation of the membrane attack complex (*Figure 7D, E*). Although the mechanisms of the *FhuB* V73A mutation in the interaction with each blood component must be further elucidated, our results indicated that the 73rd residue of FhuB is a key factor for bacterial survival in human blood during systemic infection.

Recently, He et al. reported that antibodies against the ferrichrome-binding protein FtsB, also known as FhuD, decrease blood bacterial burden and skin abscess formation in murine models infected with *emm*1 *S. pyogenes* (*He et al., 2022*). FhuD is a lipoprotein that has been studied as a potential vaccine candidate against *Staphylococcus aureus* (*Nguyen and Götz, 2016*). The detailed function and effect of the FhuBCCDG complex in *S. pyogenes* remain unknown; however, at least in *emm*1 strains, the complex can be expressed and localized abundantly on the cell membrane. Hence, this complex may be a promising target for vaccines against *S. pyogenes*.

Our gene-based pan-GWAS revealed no significant correlation with the distribution of genes encoding virulence factors. This finding minimizes the possibility that invasive infections are the result of the acquisition of virulence factors by non-invasive strains and indirectly supports the hypothesis that changes in gene expression profiles cause invasive infections. Although no vaccine has been commercialized against *S. pyogenes*, given the possibility that the pathogen has multiple gene expression patterns, even within a single lineage, it may be difficult to develop a universal vaccine with a single antigen, and a vaccine containing multiple antigens may be effective.

In the global cohort, 169 genes were related to the phenotype, including 37 phage- and 11 MGE-related genes. The genome of *S. pyogenes* is rich in prophages, phage-like chromosomal islands, and MGEs (*Bessen et al., 2015*). Prophages and MGEs sometimes function as vectors of virulence factors and antibiotic-resistance genes through interspecies and intraspecific transmission. Common superantigen-coding genes, such as *speA*, *speC*, *speG*, *speH*, *speI*, *speJ*, *speK*, and *speL*, and DNase-coding genes, such as *spd1*, are exogenous genes derived from prophages (*McShan et al., 2019*). In addition, in the genome of *S. pyogenes*, there are broad MGEs where *erm* and *mefA* related to macrolide resistance and *tetM* involved in resistance to tetracyclines are located (*Berbel et al., 2021*). In the present study, we observed no direct association between invasiveness and virulence of genes located in the prophages and MGEs. On the other hand, a cluster comprising 49 non-invasiveness-associated genes including antibiotic-resistance genes was identified. Furthermore, among the genes showing a significant correlation with the infectious phenotype, approximately 90% (152 of 169) were associated with non-invasiveness. One possible explanation is that significantly related genes reflect the process of not only gain of factors but also loss of those affecting fitness cost. Given that we also identified hypothetical genes associated with this phenotype, we assumed that there is an unknown mechanism contributing to the pathogenesis of invasive infections.

In the present study, MLST analysis revealed that 522 of the 666 strains belonged to ST101. Following ST101, ST646 was the second most prevalent lineage in the Japanese cohort and was not detected in any other country. In addition, the difference between ST101 and ST646 was found only at the 295th nucleotide in *murI*, among the seven genes subjected to MLST typing. Our phylogenetic trees also suggested that ST101 and ST646 have a close phylogenetic relationship. Five previous studies have described ST646 in *S. pyogenes*, and all of them reported *emm*89 ST646 strains isolated in Japan (*Wajima et al., 2014*; *Hasegawa et al., 2017*; *Tatara et al., 2020*; *Tanaka et al., 2016*; *Ubukata et al., 2020*). Furthermore, all ST646 strains were isolated from the 4249 *emm*89 *S. pyogenes* strains in the MLST database PubMLST (*Jolley et al., 2018*). Therefore, ST646 appears to be specific

to *emm*89 strains and a unique lineage in Japan. Notably, Ubukata et al. also reported that ST646 strains began increasing after 2012, whereas the ST101 lineage was dominant until 2008 (***Ubukata et al., 2020***). Taken together, we conclude that ST646 is possibly a relatively new and Japan-specific lineage of *emm*89 *S. pyogenes*.

Our study had three limitations. First, because we sequenced bacterial genomes using short-read sequencing, we could not detect large bacterial genome rearrangements by comparing complete genome sequences. Thus, we were unable to investigate the effects of long genomic structural dynamics, such as inversions, on pathogenesis. Second, the clinical information associated with the isolates was limited; therefore, our analyses did not reflect host information, such as age, sex, and underlying health conditions. Third, we only used bacterial genetic distances based on core genome sequences as covariates in the pan-GWAS. A combined GWAS of the host and pathogen in *S. pyogenes* infection would highlight the relationship between host risk factors and bacterial genetic variants.

In this study, we revealed the genotype–phenotype associations found in not only the known factors represented by *covS* but also factors that are related to invasiveness, including *fhuB*. Moreover, we experimentally validated the contribution of the *fhuB* mutation to bacterial survival in human blood. This study demonstrates the potential of our genomic statistical approach for elucidating the pathogenesis of invasive infections. Further analyses of the invasiveness-related factors identified in this study could provide a platform for establishing novel treatments and preventive strategies against invasive infections.

# Materials and methods

**Key resources table**

| Reagent type (species) or resource | Designation | Source or reference | Identifiers | Additional information |
|---|---|---|---|---|
| Strain, strain background (*Streptococcus pyogenes*) | TK02' | This paper | | Clinical isolate from severe invasive infection; wild-type reference for mutagenesis (see Materials and methods) |
| Strain, strain background (*Streptococcus pyogenes*) | TK02'-fhuB T218C | This paper | | Isogenic mutant strain carrying *fhuB* T218C point mutation (see Materials and methods) |
| Strain, strain background (*Escherichia coli*) | DH5α | Takara Bio | | Host strain for plasmid construction |
| Chemical compound, drug | Carbenicillin | Nacalai Tesque | | Used for selection in *E. coli* |
| Chemical compound, drug | Spectinomycin | Fujifilm Wako Pure Chemical Corporation | | Used for selection in *E. coli* and *S. pyogenes* |
| Chemical compound, drug | Mutanolysin | Sigma-Aldrich | | Used for bacterial lysis |
| Chemical compound, drug | Lysozyme | Fujifilm Wako Pure Chemical Co | | Used for bacterial lysis |
| Chemical compound, drug | Achromopeptidase | Fujifilm Wako Pure Chemical Co | | Used for bacterial lysis |
| Chemical compound, drug | RNase A | Promega | | Used for RNA degradation during DNA extraction |
| Commercial assay or kit | Maxwell RSC instrument | Promega | | Automated DNA extraction |
| Commercial assay or kit | Nextera XT DNA Kit | Illumina | | Library preparation for sequencing |
| Commercial assay or kit | QuantiChrom Iron Assay Kit | BioAssay Systems | | Used for intracellular ferric ion measurement |
| Software, algorithm | Fastp v0.20.1 | ***Chen et al., 2018*** | RRID:SCR_016962 | Used for quality filtering of sequencing reads |
| Software, algorithm | SKESA v2.4.0 | ***Souvorov et al., 2018*** | | Used for de novo genome assembly |

*Continued on next page*

*Continued*

| Reagent type (species) or resource | Designation | Source or reference | Identifiers | Additional information |
|---|---|---|---|---|
| Software, algorithm | MLST v2.19.0 | | | Used for multi-locus sequence typing |
| Software, algorithm | Prokka v1.14.5 | *Seemann, 2014* | RRID:SCR_014732 | Used for genome annotation |
| Software, algorithm | Roary v3.12.0 | *Page et al., 2015* | RRID:SCR_018172 | Used for pan-genome analysis |
| Software, algorithm | IQ-tree v1.6.12 | *Nguyen et al., 2015* | RRID:SCR_017254 | Used for phylogenetic tree construction |
| Software, algorithm | pyseer v1.3.4 | *Lees et al., 2018* | | Used for pan-GWAS |
| Software, algorithm | DBGWAS v0.5.3 | *Jaillard et al., 2018* | | Used for k-mer-based GWAS |
| Software, algorithm | AlphaFold v2.2.2 | *Jumper et al., 2021* | RRID:SCR_025454 | Used for protein structure prediction |
| Software, algorithm | Geneious Prime v2022.0.1 | Biomatters | RRID:SCR_010519 | Used for sequence visualization and mapping |
| Software, algorithm | STAR v2.7.0a | | RRID:SCR_004463 | Used for RNA-seq read mapping |
| Software, algorithm | featureCounts v1.5.2 | *Liao et al., 2014* | RRID:SCR_012919 | Used for RNA-seq read counting |
| Software, algorithm | iDEP v0.96 | *Ge et al., 2018* | | Used for transcriptomic data analysis |
| Software, algorithm | Prism v7.0c | GraphPad | RRID:SCR_002798 | Used for statistical analysis |
| Commercial assay or kit | Quick-RNA Miniprep Kit | Zymo Research | | Used for RNA extraction from bacterial samples |

## Clinical isolates in Japan

Clinical isolates were collected from public health institutions in Tokyo, Osaka, Yamaguchi, Fukushima, Kobe, Kyoto, Amagasaki, Sapporo, and Niigata, Japan. We defined the strains collected as STSS according to the Infectious Diseases Control Law in Japan. Non-invasive strains were defined based on diagnostic names, including asymptomatic, pharyngitis, tonsillitis, or non-invasive infections. Strains with no diagnostic names for the non-STSS strains were defined as non-invasive based on the isolate sites. Information on all the strains included in this study is presented in *Supplementary file 1, table S3*.

The collected *S. pyogenes* strains were cultured at 37°C in an atmosphere containing 5% $CO_2$, in Todd Hewitt broth supplemented with 0.2% yeast extract (THY; both from BD Biosciences, Franklin Lakes, NJ, USA) and stored in THY broth with 30% glycerol (Nacalai Tesque, Kyoto, Japan), at –80°C.

## Genomic DNA sequencing of the clinical isolates

The *S. pyogenes* strains were cultured until the exponential growth phase ($OD_{600}$ = 0.3–0.4). Bacterial cells were lysed with T10E1N100 buffer (10 mM Tris-HCl buffer, 1 mM EDTA, and 100 mM sodium chloride), 10 units/ml mutanolysin (Sigma-Aldrich, St. Louis, MO, USA), 10 mg/ml lysozyme (Fujifilm Wako Pure Chemical Co, Osaka, Japan), 0.5 mg/ml achromopeptidase (Fujifilm Wako Pure Chemical Co), and 0.3 mg/ml RNase A (Promega, Madison, WI, USA). Next, genomic DNA was extracted from each lysate using a Maxwell RSC instrument (Promega), according to the manufacturer's instructions, and 250 bp paired-end libraries were then generated from the extracted DNA using a Nextera XT DNA Kit (Illumina, San Diego, CA, USA). Libraries were sequenced using a NovaSeq 6000 System (Illumina) at the Genome Information Research Center, Research Institute for Microbial Diseases, The University of Osaka, Osaka, Japan. On average, the number of reads was 5,433,301 (range 3,437,124–9,117,301).

## Collection of published genome sequences

We previously sequenced the draft genomes of 161 *emm*89 clinical isolates collected in Japan between 2011 and 2019 (*Hirose et al., 2020*). We defined strains derived from STSS as 'severe invasive', and those obtained from pharyngitis, tonsillitis, and superficial skin lesions as 'non-invasive' phenotypes.

To obtain public genome sequences of *emm*89 strains isolated from other countries, we downloaded draft genome sequences in FASTA format from the National Center for Biotechnology Information (NCBI) database, using Fasterq-dump v.2.9.6. The phenotype of each strain, whether invasive or non-invasive, was defined according to the definitions in the respective references that reported the strains (*Beres et al., 2017*; *Chochua et al., 2017*; *Davies et al., 2019*).

## Genomic data processing and pan-genome analysis

All processes and analyses were performed using the National Institute of Genetics (NIG) supercomputer and SQUID at the D3 center of The University of Osaka (Osaka, Japan). We constructed a

workflow for bacterial pan-GWAS and other bioinformatic processes (*Figure 1—figure supplement 3*). All collected sequences were subjected to quality checks using Fastp v.0.20.1, with a default cutoff value of $Q > 15$ (*Chen et al., 2018*). For newly isolated strains in Japan, *emm* typing was performed using the emm-typing-tool v.0.0.1, and only sequences of strains determined as *emm*89 were used for the following analyses (*Kapatai et al., 2017*). All *emm*89 sequence data were then subjected to de novo assembly using SKESA v.2.4.0, with default parameters (*Souvorov et al., 2018*). Next, the MLST of each sequence was performed using MLST v.2.19.0 (*Jolley and Maiden, 2010*, *Maiden et al., 2013*). Clade typing was performed using BLAST v.2.13.0, with reference to the three *nga* promoter region sequences (*Zhu et al., 2015b*). After the genes were annotated with Prokka v.1.14.5, the pan-genome of all sequences was calculated using Roary v.3.12.0, with the parameters '-e -mafft -r -qc -cd 99' (*Seemann, 2014*, *Page et al., 2015*). Roary generated a core gene alignment and the distribution of all genes among the strains. To extract SNPs/indels from core genes, including single-nucleotide indels, snp-sites v.2.5.1 with the option '-v' was used. The output files were further processed using BCFtools v.1.9, with the parameter 'norm -m –', enabling analysis of multiple alleles in the pan-GWAS. In parallel, k-mers were extracted using DBGWAS v.0.5.3, and the length of k-mers was set as 31 nt using the '-k 31' parameter of DBGWAS (*Jaillard et al., 2018*).

## Phylogenetic analysis

Phylogenetic relationships were calculated from the core gene alignment, using IQ-tree v.1.16.12 (*Nguyen et al., 2015*), based on maximum likelihood. The substitution model was automatically selected considering the Akaike's and Bayesian information criteria by setting the '-m MFP' parameter of IQ-tree (*Kalyaanamoorthy et al., 2017*). Phylogenetic trees were constructed using iTOL v.6.6 (*Letunic and Bork, 2021*). The similarity of clustering in the two phylogenetic trees was statistically examined using Pearson's chi-square test with R v.4.0.3 (*R Development Core Team, 2022*), followed by post hoc analysis using residual analysis adjusted with the Holm's method, if p < 0.05.

## GWAS

To investigate the associations between phenotypes and genotypes, including SNPs/indels and genes, we performed a pan-GWAS using pyseer v.1.3.4 (*Lees et al., 2018*). The VCF file of SNPs/indels or the gene distribution matrix was designated as the genotype. To remove biases derived from lineages, we added information on the genetic distances between all pairs of strains as covariates using mash v.2.3 (*Ondov et al., 2016*). Briefly, de novo assemblies were compressed through conversion into minimum hash values using the command 'mash sketch -s 10000'. Subsequently, the commands 'mash dist' and 'square_mash' were utilized to generate a genetic distance matrix expressed with Jaccard coefficients (*Ondov et al., 2016*). The obtained matrix underwent eigenvalue decomposition, and the number of eigenvalues used for multidimensional scaling was visually determined using the plot of the relationships between the eigenvalues and contribution ratios. The number of eigenvalues and the distance matrix were then added as pyseer parameters. The pyseer calculation was iterated 1000 times with randomized phenotypes, and the 5-percentile value of the minimal p-value in each calculation was set at the significance level. Using R and the package ggplot2, the results were visualized as a Manhattan plot for the SNP/indel-based pan-GWAS and a volcano plot for the gene-based pan-GWAS, respectively (*Wickham, 2016*). Heatmaps of the strains possessing significant variants were generated using Excel (v.16.66.1; Microsoft, Redmond, WA, USA). The correlation of the presence of significant COGs was calculated and visualized using the R program.

The k-mers-based GWAS was carried out using DBGWAS (*Jaillard et al., 2018*). K-mers were considered significant at a false discovery rate (*q*-value) of <0.05. DBGWAS-calculated complexes, which are regions encompassing the k-mers, were significantly related to pathology, and de Bruijn graphs were generated based on these complexes. The sequences of the k-mers were outputted and mapped to a reference sequence using Geneious Prime v.2022.0.1 (Biomatters, Auckland, New Zealand) to identify the mutations indicated by the k-mers. For the reference strain, we adopted MGAS27061, which was isolated from an invasive case in the USA and whose complete chromosomal sequence has been used as the reference sequence of *emm*89 clade 3 (*Beres et al., 2016*). Promoter sequences in intergenic regions were predicted using web-based tools, MLDSPP and BacPP (*Paul et al., 2024*, *de Avila E Silva et al., 2011*). Additionally, BLAST was employed to search the promoter

sequences of *S. pyogenes* strain SF370 registered in the CDBProm database (https://aw.iimas.unam.mx/cdbprom/) (*Martinez et al., 2024*).

## Protein structure prediction

Significant variants found in the pan-GWAS, resulting in non-synonymous substitutions in proteins, were searched by converting nucleotide sequences into amino acid sequences using EMBOSS Transeq v.6.6.0.0 (*Rice et al., 2000*). To assess whether these mutations affected the protein function, protein structure prediction models were constructed using AlphaFold v.2.2.2 (*Jumper et al., 2021*). The calculations were performed five times for each model, by setting the option multimer_predictions_per_model = 5. We predicted multimer models using the option `--model_preset="multimer"` if a protein is reported or anticipated to form a multimer. For each monomer, we selected the model with the best predicted local difference distance test (an indicator of local structural accuracy) score (*Mariani et al., 2013*). For each multimer, AlphaFold was calculated and expressed as a weighted combination of the interface-predicted TM and predicted TM scores (ipTM + pTM). pTM is a metric for overall topological accuracy and ipTM is used to measure the structural accuracy of the protein-protein interface (*Yin et al., 2022*). The transmembrane regions of the proteins were predicted using SOSUI (https://harrier.nagahama-i-bio.ac.jp/sosui/mobile/; *Hirokawa et al., 1998*). The structures of the obtained model were visualized using PyMOL v.2.5 (Schrödinger, LLC, New York, NY, USA).

## Construction of the *fhuB* T218C mutant strains

We used the several times-passaged *S. pyogenes* TK02 strain, TK02′, as the WT strain. The TK02 strain was originally isolated from a patient with severe invasive infection (*Hirose et al., 2020*). The whole genome of TK02′ was sequenced, and the generated fasta file is available in Document S1. A point mutation, *fhuB* T218C, was introduced using the temperature-sensitive shuttle vector, pSET4s, as reported previously (*Takamatsu et al., 2001*). Sanger sequencing confirmed the presence of the point mutation. In addition, we resequenced the draft genome to confirm that there were no differences apart from the point mutation, as described above.

The bacterial strains, primers, and plasmids used in this study are listed in *Supplementary file 1, tables S15 and S16*. *Escherichia coli* strain DH5α (Takara Bio, Shiga, Japan) was used as a host for the plasmid derivatives. All *E. coli* strains were cultured in Luria Bertani broth, at 37°C, with agitation. For selection and maintenance of mutants, antibiotics were added to the media at the following concentrations: carbenicillin (Nacalai Tesque), 100 µg/ml for *E. coli*; and spectinomycin (Fujifilm Wako Pure Chemical Corporation), 100 µg/ml for *E. coli* and *S. pyogenes*.

## Transcriptomic analysis

The *fhuB* WT and mutant strains were harvested in 30 ml of THY broth, of which 1 ml was dispensed into 10 ml of THY and the remainder was centrifuged and resuspended in 2 ml of heparinized human blood. Bacterial mixtures with THY or blood were dispensed into three aliquots (*n* = 3 for each condition) and incubated at 37°C for 3 hr. THY samples were centrifuged and resuspended in RNA Shield (Zymo Research, Irvine, CA, USA). For each blood sample, 2 volumes (1 ml) of RNA protection bacteria reagent (QIAGEN, Hilden, Germany) were added. L5, included in the PureLink Total RNA Blood Purification Kit (Thermo Fisher Scientific, Waltham, MA, USA), was added to remove erythrocytes. The bacterial cell wall was mechanically lysed in Lysing Matrix B using a MagNA Lyser (Roche, Basel, Switzerland). After centrifugation, the total bacterial RNA was extracted using a Quick-RNA Miniprep Kit (Zymo Research), according to the manufacturer's instructions. Full-length cDNA was generated using the SMART-Seq HT Kit (Takara Bio), according to the manufacturer's instructions. Pair-end libraries were generated using a Nextera XT DNA Kit and sequenced using a NovaSeq 6000 system (both from Illumina, San Diego, CA, USA). Sequenced data were preprocessed using Trimmomatic v.0.33 and FastQC v.0.12.1. The reads were mapped to the complete MGAS27061 genome (NCBI reference sequence: NZ_CP013840.1) using STAR v.2.7.0a. After a second quality check using FastQC, read counting was performed using featureCounts v.1.5.2 (*Liao et al., 2014*). Differentially expressed genes were identified using iDEP v.0.96 and gene annotations from NCBI and Prokka were combined (*Ge et al., 2018*). Plots were created using iDEP and the R package ggplot2.

## Intracellular ferric ion assay

Human plasma was obtained through centrifugation of heparinized human blood, after 30 min of incubation at 37°C. The WT and *fhuB* mutant strains were harvested at the exponential phase, resuspended into 1 ml of THY or serum, and then incubated at 37°C for 3 hr. Viable bacterial cells were counted as colony-forming units (CFUs) by plating the diluted samples onto THY agar plates. Intracellular ferric ions were measured using a QuantiChrom Iron Assay Kit (BioAssay Systems, Hayward, CA, USA), according to the manufacturer's instructions. Briefly, 50 µl of standards or samples in 96-well plates were mixed with 200 µl QuantiChrom Working Reagent and incubated at 20–24°C for an hour. The optical density at the wavelength of 600 nm was measured using an Infinite 200 Pro F Plex Instrument (TECAN, Männedorf, Switzerland). The assay was repeated three times, and the results of the respective experiments were combined. Statistical analyses were performed using the Mann–Whitney *U* test.

## Bactericidal assay

The bactericidal assay was performed as described previously, with minor modifications (*Terao et al., 2008*; *Lancefield, 1957*; *Yamaguchi et al., 2019*; *Takemura et al., 2022*). Briefly, whole blood was collected from healthy adults. Human neutrophils and erythrocytes were prepared using PolymorphPrep (Serumwerk Bernburg, Bernburg, Germany), according to the manufacturer's instructions. Heparinized human blood was centrifuged at 500 × *g* for 30 min to isolate erythrocytes and polymorphonuclear cells, which were then suspended in Roswell Park Memorial Institute (RPMI)-1640 medium containing L-glutamine and Phenol Red (Fujifilm Wako Pure Chemical Corporation). Heat-inactivated plasma was prepared at 56°C for 30 min. Subsequently, 195 µl of heparinized human whole blood, erythrocytes in RPMI-1640, polymorphonuclear leukocytes in RPMI-1640, plasma, heat-inactivated plasma, or brain heart infusion broth (BD Biosciences), and 5 µl of early exponential phase bacteria with $0.9–2.0 \times 10^4$ CFUs/well were mixed in 96-well plates and incubated at 37°C, in an atmosphere containing 5% $CO_2$, for 1, 2, and 3 hr. Viable bacterial cells were counted as CFUs by plating the diluted samples onto THY agar plates. The growth index was calculated as the number of CFUs at a specified time point divided by the number of CFUs in the initial inoculum. The assay was repeated three times, and the results of the respective experiments were combined. Statistical analyses were performed using the Mann–Whitney *U* test. Differences were considered statistically significant at $p < 0.05$, using Prism v.7.0c (GraphPad, La Jolla, CA, USA).

## Ethical approval

Studies involving human participants were reviewed and approved by the Institutional Review Board of Osaka University Graduate School of Dentistry (approval nos. H26-E43 and H29-E16-2). The donors provided written informed consent to participate in the human blood bactericidal assay. For the *S. pyogenes* collection, as we retrospectively obtained clinical isolates of *S. pyogenes*, we utilized an opt-out consent procedure instead of obtaining written informed consent from the patients.

## Acknowledgements

We would like to thank the NGS core facility of the Genome Information Research Center at the Research Institute for Microbial Diseases of The University of Osaka for their support in the DNA sequencing and data analysis and the Bioinformatic Research Unit of Graduate School of Dentistry, The University of Osaka for their support in the bioinformatics analysis. This study was partially performed on the National Institute of Genetics (NIG) supercomputer at the Research Organization of Information and Systems National Institute of Genetics. This study was partly completed using SQUID at the D3 center, The University of Osaka, Japan, under the 'Joint Usage/Research Center for Interdisciplinary Large-scale Information Infrastructures (JHPCN)' (Project ID: EX22701, jh230035, jh240003, and jh250016). Masayuki Ono and Kotaro Higashi were recipients of the Iwadare Scholarship from the Iwadare Scholarship Foundation. We wish to express our gratitude to Mami Tateshita (Sapporo City Institute of Public Health) as well as the medical institutions that participated in the collection of clinical strains. This study was partly supported by AMED (JP17fk0108044, JP20fk0108130, JP20wm0325001, and JP243fa727001h), the Japan Society for the Promotion of Science KAKENHI (grant numbers 20KK0210, 22H03262, 22K19618, 22K19619, 23H03073, 23K19687, and 24K19854),

the Takeda Science Foundation, Naito Foundation, and Joint Research Program of the Research Center for GLOBAL and LOCAL Infectious Diseases, Oita University (2022B05, 2025B01, and 2025B13). This study was conducted as part of 'The Nippon Foundation – Osaka University Project for Infectious Disease Prevention.' This study was supported by JST SPRING (grant number JPMJSP2138). The funders had no role in the study design, data collection or analysis, decision to publish, or preparation of the manuscript.

## Additional information

### Funding

| Funder | Grant reference number | Author |
| --- | --- | --- |
| Japan Society for the Promotion of Science | 20KK0210 | Masaya Yamaguchi<br>Yujiro Hirose<br>Shigetada Kawabata |
| Japan Society for the Promotion of Science | 22H03262 | Masaya Yamaguchi<br>Yujiro Hirose<br>Tomoko Sumitomo<br>Shigetada Kawabata |
| Japan Society for the Promotion of Science | 22K19618 | Masaya Yamaguchi<br>Tomoko Sumitomo<br>Shigetada Kawabata |
| Japan Society for the Promotion of Science | 22K19619 | Masaya Yamaguchi<br>Shigetada Kawabata |
| Japan Society for the Promotion of Science | 23H03073 | Masaya Yamaguchi<br>Tomoko Sumitomo<br>Shigetada Kawabata |
| Japan Society for the Promotion of Science | 23K19687 | Masayuki Ono |
| Japan Society for the Promotion of Science | 24K19854 | Masayuki Ono |
| Joint Research Program of the Research Center for GLOBAL and LOCAL Infectious Diseases, Oita University | 2022B05 | Shigetada Kawabata |
| Joint Research Program of the Research Center for GLOBAL and LOCAL Infectious Diseases, Oita University | 2025B01 | Masaya Yamaguchi |
| Joint Research Program of the Research Center for GLOBAL and LOCAL Infectious Diseases, Oita University | 2025B13 | Masayuki Ono |
| Japan Science and Technology Agency SPRING | JPMJSP2138 | Masayuki Ono<br>Kotaro Higashi |
| AMED | JP17fk0108044 | Tadayoshi Ikebe<br>Shigetada Kawabata |
| AMED | JP20fk0108130 | Tadayoshi Ikebe<br>Shigetada Kawabata |
| AMED | JP20wm0325001 | Masaya Yamaguchi<br>Shigetada Kawabata |
| AMED | JP243fa727001h | Masaya Yamaguchi |

| Funder | Grant reference number | Author |
| --- | --- | --- |
| Joint Usage/Research Center for Interdisciplinary Large-scale Information Infrastructures (JHPCN) | EX22701 | Masaya Yamaguchi |
| Joint Usage/Research Center for Interdisciplinary Large-scale Information Infrastructures (JHPCN) | jh230035 | Masaya Yamaguchi |
| Joint Usage/Research Center for Interdisciplinary Large-scale Information Infrastructures (JHPCN) | jh240003 | Masaya Yamaguchi |
| Joint Usage/Research Center for Interdisciplinary Large-scale Information Infrastructures (JHPCN) | jh250016 | Masaya Yamaguchi |

The funders had no role in study design, data collection, and interpretation, or the decision to submit the work for publication.

## Author contributions

Masayuki Ono, Data curation, Formal analysis, Funding acquisition, Validation, Investigation, Visualization, Methodology, Writing – original draft, Writing – review and editing, Conceptualization; Masaya Yamaguchi, Conceptualization, Data curation, Supervision, Methodology, Writing – review and editing, Formal analysis, Funding acquisition, Investigation, Project administration, Validation, Writing – original draft; Daisuke Motooka, Resources, Investigation, Writing – review and editing, Formal analysis, Methodology; Yujiro Hirose, Resources, Writing – review and editing, Methodology; Kotaro Higashi, Investigation, Visualization, Writing – review and editing, Funding acquisition, Methodology; Tomoko Sumitomo, Writing – review and editing, Methodology, Resources; Tohru Miyoshi-Akiyama, Rumi Okuno, Takahiro Yamaguchi, Ryuji Kawahara, Hitoshi Otsuka, Noriko Nakanishi, Yu Kazawa, Chikara Nakagawa, Ryo Yamaguchi, Hiroo Sakai, Yuko Matsumoto, Resources, Writing – review and editing; Tadayoshi Ikebe, Resources, Writing – review and editing, Funding acquisition; Shigetada Kawabata, Conceptualization, Supervision, Project administration, Writing – review and editing, Funding acquisition

## Author ORCIDs

Masayuki Ono ⬤ https://orcid.org/0009-0006-9557-9669
Masaya Yamaguchi ⬤ https://orcid.org/0000-0001-6218-7112
Yujiro Hirose ⬤ https://orcid.org/0000-0001-6338-4767
Kotaro Higashi ⬤ https://orcid.org/0000-0001-9529-1557
Tomoko Sumitomo ⬤ https://orcid.org/0000-0003-0653-0339

Reviewer #1 (Public review): https://doi.org/10.7554/eLife.101938.3.sa1
Author response https://doi.org/10.7554/eLife.101938.3.sa2

# Additional files

## Supplementary files

MDAR checklist

Supplementary file 1. Tables S1–S16.

Source data 1. Sequencing data for the *fhuB* T218C mutant strain.

## Data availability

Data for the 207 sequenced *S. pyogenes* genomes were deposited in the DDBJ sequence read archive, under BioProject PRJDB16457. The DRR run number is DRR511668-DRR511874.Sequencing data for the *fhuB* T218C mutant strain is provided as Supplementary Data 1.

The following datasets were generated:

| Author(s) | Year | Dataset title | Dataset URL | Database and Identifier |
|---|---|---|---|---|
| Ono M, Yamaguchi M, Kawabata S | 2025 | Illumina NovaSeq 6000 paired end sequencing of KB01 | https://www.ncbi.nlm.nih.gov/sra/DRR511668 | NCBI Sequence Read Archive, DRR511668 |
| Ono M, Yamaguchi M, Kawabata S | 2025 | llumina NovaSeq 6000 paired end sequencing of TK124 | https://www.ncbi.nlm.nih.gov/sra/DRR511874 | NCBI Sequence Read Archive, DRR511874 |

The following previously published datasets were used:

| Author(s) | Year | Dataset title | Dataset URL | Database and Identifier |
|---|---|---|---|---|
| Hirose Y, Yamaguchi M, Takemoto N, Miyoshi-Akiyama T, Sumitomo T, Nakata M, Ikebe T, Hanada T, Yamaguchi T, Kawahara R | 2020 | Genetic characterization of recently emerged clade of emm 89 *Streptococcus pyogenes*, Japan, 2011-2019 | https://www.ncbi.nlm.nih.gov/bioproject/PRJDB8877 | NCBI BioProject, PRJDB8877 |
| Chochua S, Metcalf BJ, Li Z, Rivers J, Mathis S, Jackson D, Gertz RE, Srinivasan V, Lynfield R, Van Beneden C | 2017 | WGS of the United States invasive *Streptococcus pyogenes* isolates | https://www.ncbi.nlm.nih.gov/bioproject/PRJNA395240 | NCBI BioProject, PRJNA395240 |
| Davies MR, McIntyre L, Mutreja A, Lacey JA, Lees JA, Towers RJ, Smeesters PR, Frost HR, Price DJ, Holden MT G | 2019 | Complete genomes of 30 globally distributed Group A Streptococcus isolates | https://www.ncbi.nlm.nih.gov/bioproject/PRJNA454341 | NCBI BioProject, PRJNA454341 |
| Beres SB, Olsen RJ, Saavedra MO, Ure R, Reynolds A, Lindsay DSJ, Smith AJ, Musser JM | 2017 | Scotland *Streptococcus pyogenes* emm89 strains, 2010-2016 | https://www.ncbi.nlm.nih.gov/bioproject/PRJNA387243 | NCBI BioProject, PRJNA387243 |

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
