## [Editor Report · eLife Assessment]

This study provides an **important** and timely analysis of invasive and non-invasive *Streptococcus pyogenes emm*89 isolates, which have become a dominant serotype in the past decade. Using genome sequencing of 311 strains from Japan and comparing them with 666 global strains, the authors present **compelling** evidence in support of the identification of genetic factors linked to the invasive phenotype of *emm*89. The findings are both theoretically and practically significant in medical microbiology.

---

## [Referee Report · Reviewer #1 (Public review)]

Summary:

In this study, the authors sequenced emm89 serotype genomes of clinical isolates from patients in Japan, where the number of invasive Group A Streptococcus (GAS), especially those of the emm89 serotype, has drastically increased over the past 10-15 years. The sequences from this cohort were compared against a large collection of publicly available global isolates, yielding a total of almost 1000 genomes in the analysis. Because the researchers focused on the emm89 serotype, they could construct a common core genome, with subsequent ability to analyze genomic differences in accessory genes and intergenic regions that contributed to the invasive phenotype using multiple types of GWAS analysis (SNP, k-mer). Their analysis demonstrates some mutations responsible for invasiveness are specific to the Japanese strains, and that multiple independent virulence factors can contribute to invasiveness. None of the invasive phenotypes were correlated with new gene acquisition. Together, the data support that synergy between bacterial survival and upregulation of virulence factors contribute to the development of severe infection.

Strengths:

• The authors verify their analysis by confirming that covS is one of the more frequently mutated genes in invasive strains of GAS, as has been shown in other publications.

• A mutation in one of the SNPs attributed to invasiveness (SNP fhuB) was introduced into an invasive strain. The authors demonstrate that this mutant strain survives less well in human blood. Therefore, the authors have experimental data to support their claims that their analysis uncovered a new mutation/SNP that contributed to invasiveness.

Weaknesses:

• It would be helpful for the authors to highlight why their technique (large scale analysis of one emm type) can yield more information than a typical GWAS analysis of invasive vs. non-invasive strains. Are SNPs easier to identify using a large-scale core genome? Is it more likely evolutionarily to find mutations in non-coding regions as opposed to the core genome and accessory genes, and this is what this technique allows? Did the analysis yield unexpected genes or new genes that had not been previously identified in other GWAS analyses? These points may need to be made more apparent in the results and deserves some thought in the discussion section.

• The Alpha-fold data does not demonstrate why the mutations the authors identified could contribute to the invasive phenotype. It would be helpful to show an overlay of the predicted structures containing the different SNPs to demonstrate the potential structural differences that can occur due to the SNP. This would make the data more convincing that the SNP has a potential impact on the function of the protein. Similarly, the authors discuss modification of the hydrophobicity of the side chain in the ferrichrome transporter (lines 317-318) due to a SNP, but this is not immediately obvious in the figure (Fig. 5).

Comments on revisions:

The authors have addressed the concerns from reviewers. The implemented revisions have improved the manuscript's clarity.

---

## [Author Response]

The following is the authors’ response to the original reviews

**Reviewer #1 (Public review):**
Weaknesses:It would be helpful for the authors to highlight why their technique (large scale analysis of one emm type) can yield more information than a typical GWAS analysis of invasive vs. non-invasive strains. Are SNPs easier to identify using a large-scale core genome? Is it more likely evolutionarily to find mutations in non-coding regions as opposed to the core genome and accessory genes, and this is what this technique allows? Did the analysis yield unexpected genes or new genes that had not been previously identified in other GWAS analyses? These points may need to be made more apparent in the results and deserve some thought in the discussion section.

We thank the reviewer for pointing out the importance of this study. By focusing on bacteria within a single *emm* type, false positives caused by confounding lineage effects can be minimized, which contributes to greater accuracy of the pan-GWAS. We have added relevant text describing the strong points of our pan-GWAS approach to the Results and Discussion sections, as shown following:

“The present pan-GWAS of bacteria within a single *emm* type minimized lineage effects, thus reducing false positives.” (lines 204–205)

The present study focused on *emm*89 *S. pyogenes*, known to cause increasing rates of invasive infections worldwide, and also assessed differences between *emm*89 strains causing invasive and non-invasive infections. By focusing on bacteria within closed phylogenies, false positives caused by confounding lineage effects were minimized, thus contributing to a higher level of accuracy of the pan-GWAS.” (lines 420–424)

In addition, we would like to comment more regarding the reviewer’s question, "Is it more likely evolutionarily to find mutations in non-coding regions as opposed to the core genome and accessory genes, and this is what this technique allows?". Mutations are generally considered to be more frequent in non-coding than coding regions. However, the actual mutation frequencies in both types of regions were not assessed in this study. Nevertheless, exploring non-coding regions using the k-mer method is of considerable importance, as variations significantly associated with infectious phenotypes may contribute to alterations in gene expression and other regulatory mechanisms.

The Alpha-fold data does not demonstrate why the mutations the authors identified could contribute to the invasive phenotype. It would be helpful to show an overlay of the predicted structures containing the different SNPs to demonstrate the potential structural differences that can occur due to the SNP. This would make the data more convincing that the SNP has a potential impact on the function of the protein. Similarly, the authors discuss modification of the hydrophobicity of the side chain in the ferrichrome transporter (lines 317-318) due to a SNP, but this is not immediately obvious in the figure (Fig. 5).

As the reviewer suggested, we have substituted Figure 5E in the previous version with a figure illustrating the molecular surface within proximity of the mutation. We speculated that the mutation may induce a small indentation on the surface, and thus attenuate the stability of the hydrophobic bound between FhuB and FhuD by invasion of solvent into the indentation. Additionally, images showing the wild-type and mutated models have been separated for better visibility instead of as an overlay of the predicted models suggested by the reviewer. Relevant text in the Results section and legend of Figure 5E have been accordingly revised, as shown following:

“The mutation was predicted to induce formation of a small indentation on the molecular surface, thus increasing the surface area accessible to the solvent, and is considered to potentially affect the stability of the hydrophobic bond between FhuB and FhuD, and thus ferrichrome transport (Figure 5E).” (lines 360–363)

“The 73rd valine in FhuB, shown in magenta, was substituted with alanine. The molecular surface is illustrated with a wireframe and that of the predicted indentation is shown with an arrowhead.” (lines 1162–1164)

**Reviewer #1 (Recommendations for the author):**
The figure legend for Fig. 3C needs to be explained so that it is similarly laid out as in Fig. 2C. Fig. 2C should indicate that the magenta color represents the invasive phenotype.

Based on this helpful suggestion, more detailed information about the magenta color representing the invasive phenotype has been added to the legends of Fig. 2C and 3C, with relevant text also included in the revised legends, as shown following:

“Colored bars above indicate countries and phenotypes, and magenta bars represent invasive phenotypes. Using the Roary program, gene names starting with “Group_” were automatically assigned. Position indicates the location of each SNP/indel on the core gene alignment. The full results are shown in Table S6.” (lines 1116–1120)

“Colored bars above indicate countries and phenotypes, and magenta bars represent invasive phenotypes. Using the Roary program, gene names starting with “Group_” were automatically assigned. The full results are shown in Table S8. (lines 1130–1133)

The wording and organization of results in the k-mer section started to get confusing around lines 270-271. It begins to be a list of results and would be better served by some interpretation or explanation of the significance (why it is important to find such mutations). For example, for mutations you find in non-coding regions, do you expect them to have a detrimental effects on gene expression/regulation?

As the reviewer kindly suggested, we have added interpretation or explanation of the significance of Comp_6 and Comp_24 to the Results section. We analyzed the function of the non-coding region of Comp_6 by employing web-based *in silico* tools, including MLDSPP and BacPP, though no promoter sequences could be identified. Next, using BLAST, a search for known promoter sequences of *S. pyogenes* M1 strain SF370 of the CDBProm database was attempted, because the web-based *in silico* promoter prediction tools are not suitable for *S. pyogenes*. However, neither identical nor homologous sequences were detected. Thus, the significance of this region remains unknown. In Comp_24, *group_141* was also identified in the COGs-based pan-GWAS as a non-invasiveness related gene. Furthermore, *group_141* showed high levels of correlation with *group_139* and *group_467*, encoding transposase and uncharacterized protein, respectively, which suggests that the presence of an MGE is associated with a non-invasive phenotype.

Relevant text has been added to the Materials and Methods (lines 653–657) and Results (lines 308–311 and 314–319) sections, as shown following:

“Promoter sequences in intergenic regions were predicted using web-based tools, MLDSPP and BacPP[29,30]. Additionally, BLAST was employed to search the promoter sequences of *S. pyogenes* strain SF370 registered in the CDBProm database (https://aw.iimas.unam.mx/cdbprom/)[69]” (lines 653–657)

“We speculated that this region is related to regulation of gene expression. However, no promoter sequences were identified by utilizing MLDSPP, BacPP, and BLAST, thus the significance of this region remains to be clarified[29,30].” (lines 308–311)

“Furthermore, *group_141* was also identified in the COGs-based pan-GWAS as a non-invasiveness-related gene along with *group_139* and *group_467*, which encode transposase and uncharacterized protein, respectively (Table S8 and Figure S4). Taken together, the absence of an MGE containing *group_141*, and the presence of another MGE harboring *group_142* and *group_143* may result in an invasive phenotype.” (lines 314–319)

Additionally, new references (#29, 30, and 69) concerning bacterial promoter prediction have been included in the revised version of the manuscript.

Because there is no difference in intracellular free ferric ions in the fhuB mutant compared with the wild-type, the authors speculate that the upregulation of the fhuBCD operon can compensate for the loss of function of the fhuB gene, but there is insufficient data to support this claim.

As the reviewer indicate, the data presented in the previous version were insufficient to support our speculation. Therefore, the following sentence has been deleted from the manuscript (previous version line 367):

“Therefore, the upregulation of *fhuBCD* may compensate for the impaired function mediated by SNP T218C.”

The authors mention that there was no direct association between invasiveness and acquisition of genes (lines 451-455), including antibiotic resistance genes from prophages and MGEs (lines 467-469). These data should be moved to the results section to focus the results on the correlation between invasiveness and mutation of existing DNA vs acquisition of new DNA.

Accordingly, we have added relevant text to the Results section, as shown following:

“On the other hand, the present pan-GWAS found no genes encoding known virulence factors significantly associated with invasiveness, thus further analysis of the relationships of detected distribution patterns with prophages and MGEs was performed.” (lines 264–267)

Minor spelling error at line 210 ("waws" instead of "was").

As the reviewer kindly pointed out, the spelling has been corrected. (line 233)

**Reviewer #2 (Recommendations for the authors):**
Minor comments:

Line 55: Does this rate apply to all types of infections?

The authors appreciate this question from the reviewer. We checked what types of infections the mortality rate is applied to and confirmed that it only represents STSS. Therefore, relevant text has been revised, as shown following:

However, even with proper treatment, the mortality rate of patients with STSS remains high, ranging from 23–81%[6]”. (lines 72–73)

Line 58: Could you explain the protein encoded by the emm gene and the role of the hypervariable region in pathogenesis?

As requested, relevant text regarding the pathogenic role of the hypervariable region of M protein has been added, as shown following:

“*S. pyogenes* has been classified into at least 240 *emm* types based on a hypervariable region sequence of the *emm* gene, which encodes the M protein. This hypervariable region of the M protein is responsible for type-specific antigenicity and binds with high affinity to C4b-binding protein, a major fluid phase inhibitor of the classical and lectin pathways of the complement system that confers resistance to opsonophagocytosis[8].” (lines 76–81)

Line 161: Figure 1C does not show the strain with the different pattern.

The authors apologize for the lack of clarity. In Fig. 1C, the strain is shown by a pale pink color bar used to indicate the related clade. For clarity, an arrowhead pointing to the strain from outside of the tree has been added along with the following text in the legend:

“Arrowhead indicates strain belonging to the novel clade.” (lines 1102–1103)

Line 239: It could be interesting to examine the genes in the region between the mobile elements found in the global cohort, as the result profile was very different from the Japanese group, which revealed more specific genes. Consider adding this to the results section.

Based on the reviewer’s insightful suggestion, we attempted to find regions between the mobile genetic element-related genes. However, contigs generated from short reads were not adequate to identify such a genome structure. Therefore, calculations to analyze the pairwise correlation of the presence of significant COGs in the 666 strains to predict genes on prophages and MGEs were performed, and the results added to Figure S4. Eight clusters were detected as coexisting COG groups, seven of which comprised phage- or MGE-related genes. Furthermore, a cluster with antimicrobial-resistant genes was shown to be correlated with non-invasive infections. It is thus speculated that gain or loss of gene sets via phages and MGEs rather than acquisition of virulence genes may lead to changes in fitness to the environment and bacterial phenotypes. Relevant text has been added to the revised versions of the Results, Discussion, and Materials and Methods sections, as shown following:

“On the other hand, the present pan-GWAS found no genes encoding known virulence factors significantly associated with invasiveness, thus further analysis of the relationships of detected distribution patterns with prophages and MGEs was performed. For calculating the pairwise correlation of the presence of significant COGs in the 666 strains, the COGs were clustered into eight coexisting groups, seven of which contained phage- and/or MGEs-related genes (Figure S4). The largest group comprised 65 genes including phage proteins, while the second largest with 42 genes was found to be associated with non-invasive infections, and included *group_2689*, *group_1833*, and *ermA1*, encoding TetR/AcrR family transcriptional regulator, multidrug efflux system permease protein, and rRNA adenine N-6-methyltransferase, respectively.” (lines 264–273)

“On the other hand, a cluster comprising 49 non-invasiveness-associated genes including antibiotic-resistance genes was identified. Furthermore, among the genes showing a significant correlation with the infectious phenotype, approximately 90% (152 of 169) were associated with non-invasiveness. One possible explanation is that significantly related genes reflect the process of not only gain of factors but also loss of those affecting fitness cost.” (lines 517–522)

“The correlation of the presence of significant COGs was calculated and visualized using the R program.” (lines 643–644)

Line 548: What cutoff values were used in Fastp?

The default cutoff value for Fastp (Q>15) was used, and relevant text has been added to the Materials and Methods section in the revised version, as shown following:

“All collected sequences were subjected to quality checks using Fastp v.0.20.1, with a default cutoff value of Q>15[53].” (lines 600–601)

Line 635: Were the transcriptome experiments performed in triplicate?

We apologize for the confusion. The transcriptome experiment was performed only once with three samples for each condition. The notation “(n=3 for each condition)” has been added to the relevant text portion in the Materials and Methods section (line 696).

Discussion section: I believe the authors should place more emphasis on the fact that FhuB is associated with non-invasiveness, to provide clearer context in the discussion.

Based on this helpful suggestion, we have revised relevant text in the Discussion section, as shown following:

“Transcriptomic analysis findings suggested that the Japan-specific *fhuB* mutation associated with non-severe invasive infections contributes to the growth rate of *S. pyogenes* in human blood by adapting to the environment.” (lines 457–459)

Also, “V73A” has been removed from the relevant text in the Discussion section to provide a more clear and precise context, with the revised sentence shown following:

“Two possible roles of the FhuB mutation in the pathogenesis of severe invasive infections are thus proposed.” (lines 470–471)